# ASSOCIATIVE MEMORY AND DEAD NEURONS

**Vladimir Fanaskov**
AIRI, Skoltech
`fanaskov.vladimir@gmail.com`

**Ivan Oseledets**
AIRI, Skoltech

## ABSTRACT

In "Large Associative Memory Problem in Neurobiology and Machine Learning," Dmitry Krotov and John Hopfield introduced a general technique for the systematic construction of neural ordinary differential equations with non-increasing energy or Lyapunov function. We study this energy function and identify that it is vulnerable to the problem of dead neurons. Each point in the state space where the neuron dies is contained in a non-compact region with constant energy. In these flat regions, energy function alone does not completely determine all degrees of freedom and, as a consequence, can not be used to analyze stability or find steady states or basins of attraction. We perform a direct analysis of the dynamical system and show how to resolve problems caused by flat directions corresponding to dead neurons: (i) all information about the state vector at a fixed point can be extracted from the energy and Hessian matrix (of Lagrange function), (ii) it is enough to analyze stability in the range of Hessian matrix, (iii) if steady state touching flat region is stable the whole flat region is the basin of attraction. The analysis of the Hessian matrix can be complicated for realistic architectures, so we show that for a slightly altered dynamical system (with the same structure of steady states), one can derive a diverse family of Lyapunov functions that do not have flat regions corresponding to dead neurons. In addition, these energy functions allow one to use Lagrange functions with Hessian matrices that are not necessarily positive definite and even consider architectures with non-symmetric feedforward and feedback connections.

## 1 INTRODUCTION

Associative or content-addressable memory is a system that retrieves the most appropriate stored pattern based on a partially known or distorted input pattern. One particularly influential realization of associative memory was proposed by John Hopfield in Hopfield (1982) for discrete variables and in Hopfield (1984) for continuous variables. Both models are distinguished by their biological plausibility, autonomy, asynchronous operations of constituent parts, robustness to noise, and strong theoretical guarantees. Later in Krotov & Hopfield (2020), it was shown that one could develop a general biologically plausible model that unites many previously known models and allows building novel associative memory systems Krotov (2023).

The model in Krotov & Hopfield (2020) is based on the nonlinear dynamical system that evolves in time from a given initial state. Nonlinear dynamical systems show an exceptionally diverse set of behavior Strogatz (2018), so one needs to select an appropriate class of ordinary differential equations suitable to model associative memory. The most crucial requirement is the ability of the system to evolve to a single state from many initial conditions that are close according to some problem-specific metrics. This fact suggests one should use a dynamical system with many stable steady states. If one selects such a system, each stable steady state corresponds to a particular memory and basin of attraction – all initial conditions that evolve to a selected state – defines a measure of similarity between states.

The technique of choice to study the stability of steady state is to construct energy or Lyapunov function Lyapunov (1992). This function, defined on the state of a dynamical system, is non-increasing on trajectories of a dynamical system. If it is possible to find such a function, its isolated local minima will correspond to steady states. Different variants of energy functions with this property are available in all previous works on associative memory Hopfield (1982), Hopfield (1984), Krotov

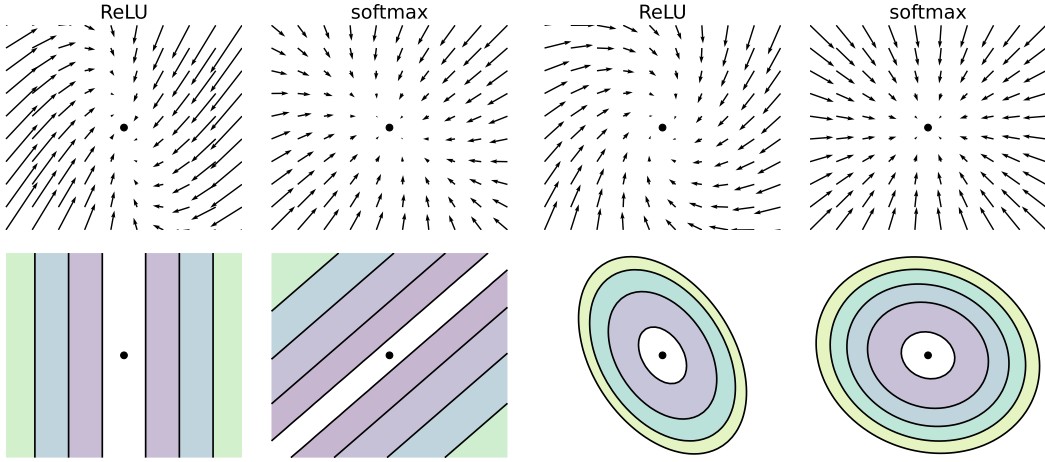

(a) Energy function from Krotov & Hopfield (2020).  (b) Energy function proposed in this article.

Figure 1: Vector fields of dynamical systems (top row) and level sets of energy functions (bottom row) for energy functions (2) (the one from Krotov & Hopfield (2020)) and (10) (proposed in this article). Vector fields suggest that all steady states are stable, but energy function (2) does not indicate that. The reason is energy function (2) has non-compact flat regions touching each point where one or more neurons die (see Proposition 1 for precise statement).

& Hopfield (2016) and recent work Krotov & Hopfield (2020) provides the most general energy function that embraces all previous ones.

Since associative memory from Krotov & Hopfield (2020) has an associated energy function, one can argue that it is a member of a broad family of energy-based models LeCun et al. (2006), including the diffusion models Hoover et al. (2023). In addition to that, the dynamical system used is clearly related to another subfield of machine learning known as Neural Ordinary Differential Equation Chen et al. (2018). Finally, in Krotov (2021), Hoover et al. (2022) it was shown that one can select parameters of model Krotov & Hopfield (2020) in such a way that dynamical variables correspond to the activation pattern of deep neural network with feedback connection. Given that, one can argue that modern Hopfield associative memory Krotov & Hopfield (2020) is uniquely tying up several major paradigms in deep learning.

In this article, we study the Lyapunov function proposed in Krotov & Hopfield (2020) and later adopted and modified in other recent papers, e.g., Hoover et al. (2024), Hoover et al. (2022), Millidge et al. (2022a). Our main observation is that energy function is vulnerable to the problem of dead neurons[1] Lu et al. (2019), which result in a flat energy direction. We illustrate this in the left panel of Figure 1, where one can find both the vector field of the dynamical system and isolines of the energy function. Clearly, the energy function from Krotov & Hopfield (2020) does not ensure stability and is not helpful in identifying a steady state when dead neurons are present.

As a remedy, we propose a slightly modified dynamical system that has no flat direction (see right panel of Figure 1) but retains other good properties of the Krotov and Hopfield model. In particular, it is still an energy-based model, neural ODE, and has steady states related to deep neural networks.

A more detailed breakdown of our contribution is below:

1. In Section 2 we provide examples of flat energy directions caused by dead neurons (Figure 1; Examples 2, 3, 4) and formally characterize architectures that are vulnerable to that problem in Proposition 1.

---

[1]Usually, neurons are called dead when activation function saturates, e.g., $x < 0$ for ReLu or $x \gg 1$ for sigmoid. In this article we use an extended definition of dead neurons and say that a neuron is dead when it is impossible to reconstruct input to activation function from its output. This is explained in more detail in Section 2.

2. Flat energy directions, as we show in Section 3, cause several undesirable consequences for sensitivity and stability:

   (a) Problems with sensitivity described in Section 3.1 include: (i) energy is completely independent of degrees of freedom corresponding to dead neurons, because of that activation functions are effectively invertible and energy function presented in Krotov & Hopfield (2020) is completely equivalent to the old one given in Hopfield (1984); (ii) In the flat regions energy function is not sensitive to the change in bias term (Section 3.1).

   (b) Stability is discussed in Section 3.2 with the following main takeaways: (i) for steady state with dead neurons, stability condition can not be ensured from the Lyapunov function alone; (ii) independent analysis of dynamical system shows dead neurons do not compromise stability, and only the motion in the range of Hessian of Lagrange function is relevant; (iii) if the steady state is stable, the whole flat direction belongs to the basin of attraction; (iv) if in addition of energy, the range of the Hessian is available (e.g., as the orthogonal projector), this information is enough to restore degrees of freedom that energy alone misses at steady state.

3. As a remedy, in Section 4, we define a slightly modified dynamical system and a family of energy functions with good properties: (i) proposition 5 shows that in general one may construct a large family of Lyapunov function with no flat directions; (ii) example 5 shows several concrete choices of Lyapunov functions for symmetric weight matrices with no restriction Hessian matrix of Lagrange function; (iii) examples 6 and 7 further relax restriction on parameters and allow one to consider memory models with non-symmetric weight matrices.

## 2 DEAD NEURONS

In Krotov & Hopfield (2020) authors proposed to rewrite activation function $\boldsymbol{g}(\boldsymbol{y})$ as gradient of Lagrange function $L(\boldsymbol{y})$, i.e., $\boldsymbol{g}(\boldsymbol{y}) = \frac{\partial L(\boldsymbol{y})}{\partial \boldsymbol{y}}$. This formalism allows to describe a large class of memory models with Lyapunov function on the common grounds. Results that we obtain in the article are applicable to all of them, but for simplicity we consider dense model taken from (Krotov, 2021, Equation (2)) (see Appendix A or further discussion on relations between different models). This model reads:

$$\dot{\boldsymbol{y}}(t) = \boldsymbol{W}\boldsymbol{g}(\boldsymbol{y}(t)) - \boldsymbol{y}(t) + \boldsymbol{b}, \; \boldsymbol{y}(0) = \boldsymbol{y}_0, \tag{1}$$

$$E(\boldsymbol{y}) = (\boldsymbol{y} - \boldsymbol{b})^\top \boldsymbol{g}(\boldsymbol{y}) - L(\boldsymbol{y}) - \frac{1}{2}\left(\boldsymbol{g}(\boldsymbol{y})\right)^\top \boldsymbol{W}\boldsymbol{g}(\boldsymbol{y}). \tag{2}$$

Equation (1) describes a temporal dynamics of feature vector $\boldsymbol{y}$ starting from $\boldsymbol{y}_0$. The equation contains weights $\boldsymbol{W}$, bias $\boldsymbol{b}$ and activation function $\boldsymbol{g}(\boldsymbol{y})$. Energy function (2) is non-increasing on trajectories of (1) if and only if $\boldsymbol{W} = \boldsymbol{W}^\top$ and Hessian of Lagrange function $\boldsymbol{\Lambda} = \frac{\partial^2 L(\boldsymbol{y})}{\partial \boldsymbol{y}^2}$ is positive semi-definite (see (Krotov & Hopfield, 2020, Equation (4), Appendix A) and (Krotov, 2021, Equation (4), Appendix A)).

Dynamical system (1) has several favorable properties (see discussion in Section 1 and Krotov & Hopfield (2020) , Krotov (2021)): (i) steady states can be used as memory vectors, (ii) dynamics in the basins of attraction naturally model memory recovery process, (iii) energy function can be used to ensure the existence of steady states and basins of attraction, (iv) memory is related to neural ODEs so can be trained end-to-end, (v) with a special choice of $\boldsymbol{W}$ steady states resemble activation pattern of classical deep learning architectures. The example below illustrates the last point (see (Krotov, 2021, Section 3)).

**Example 1** (MLP with feedback connections). *For simplicity, we consider four layers, extensions to a larger number of layers are straightforward. Weights and state vectors are partitioned on blocks of conformable size*

$$\boldsymbol{W} = \begin{pmatrix} 0 & \boldsymbol{W}_{12} & & \\ \boldsymbol{W}_{21} & 0 & \boldsymbol{W}_{23} & \\ & \boldsymbol{W}_{32} & 0 & \boldsymbol{W}_{34} \\ & & \boldsymbol{W}_{43} & 0 \end{pmatrix}, \boldsymbol{y} = \begin{pmatrix} \boldsymbol{y}_1 \\ \boldsymbol{y}_2 \\ \boldsymbol{y}_3 \\ \boldsymbol{y}_4 \end{pmatrix}, \boldsymbol{b} = \begin{pmatrix} \boldsymbol{b}_1 \\ \boldsymbol{b}_2 \\ \boldsymbol{b}_3 \\ \boldsymbol{b}_4 \end{pmatrix}.$$

*Note that $\boldsymbol{W}$ is symmetric so $\boldsymbol{W}_{12} = \boldsymbol{W}_{21}^\top$ With that choice dynamical system (1) becomes*

$$\dot{\boldsymbol{y}}_i = \boldsymbol{W}_{i,i-1}\boldsymbol{g}\left(\boldsymbol{y}_{i-1}\right) + \boldsymbol{W}_{i,i+1}\boldsymbol{g}\left(\boldsymbol{y}_{i+1}\right) - \boldsymbol{y}_i + \boldsymbol{b}_i,\ i = 2,3$$
$$\dot{\boldsymbol{y}}_1 = \boldsymbol{W}_{12}\boldsymbol{g}\left(\boldsymbol{y}_2\right) - \boldsymbol{y}_1 + \boldsymbol{b}_1,\ \dot{\boldsymbol{y}}_4 = \boldsymbol{W}_{43}\boldsymbol{g}\left(\boldsymbol{y}_3\right) - \boldsymbol{y}_4 + \boldsymbol{b}_4,$$

*so steady-state indeed resembles MLP but with feedback connections that are symmetric.*

Besides MLP described in Example 1 associative memory (1) leads to many more interesting and fruitful connections. In Krotov & Hopfield (2020) it allowed the authors to reconstruct dense associative memory Krotov & Hopfield (2016) and modern Hopfield network Ramsauer et al. (2020). In Krotov (2021) and Hoover et al. (2022) it was used to build a memory model with dense hidden layers and convolution neural networks with pooling layers. In Hoover et al. (2024) authors introduced an energy transformer using the same formalism. The authors of Tang & Kopp (2021) also noticed that MLP-mixer Tolstikhin et al. (2021) is related to associative memory (1). Clearly, the technique is valuable and versatile.

As we have seen, steady states of dynamical systems are important to emulate deep learning architectures and classical memory models. The role of the Lyapunov function is to ensure stability. Unfortunately, if one looks closer at the energy function (2), it becomes clear that it shows some pathological behavior. Before providing a formal description, we illustrate this with three examples.

**Example 2** (flat energy with ReLU activations). *Lagrange function $L(\boldsymbol{y}) = \sum_{i=1}^N \frac{1}{2}\left(ReLU(y_i)\right)^2$ corresponds to fully-connected neural network with $N$ neurons and $ReLU(x) = \frac{1}{2}\left(x + |x|\right)$ nonlinearity. Energy function becomes $E(\boldsymbol{y}) = (\boldsymbol{y} - \boldsymbol{b})^\top ReLU(\boldsymbol{y}) - \sum_{i=1}^N \frac{1}{2}\left(ReLU(y_i)\right)^2 - \frac{1}{2}\left(ReLU(\boldsymbol{y})\right)^\top \boldsymbol{W}\,ReLU(\boldsymbol{y})$. It is easy to see that if neuron $i$ dies, i.e., $y_i \leq 0$, energy becomes zero for all $\widetilde{\boldsymbol{y}} = \boldsymbol{y} + \alpha\boldsymbol{e}_i$, where $\alpha \leq 0$. $\boldsymbol{e}_i$ is a $i$-th columns of identity matrix and $\boldsymbol{y}$ is a state vector with $y_i \leq 0$. In other words, energy is non-discriminative in a non-compact region of state space.*

**Example 3** (flat energy with sigmoid activations). *For sigmoid activation function $\sigma(x) = (1 + \exp(-x))^{-1}$ Lagrange function reads $L(\boldsymbol{y}) = \sum_i \log\left(1 + e^{y_i}\right)$, and the energy is $E(\boldsymbol{y}) = (\boldsymbol{y} - \boldsymbol{b})^\top \boldsymbol{\sigma}(\boldsymbol{y}) - \sum_i \log\left(1 + e^{y_i}\right) - \frac{1}{2}\left(\boldsymbol{\sigma}(\boldsymbol{y})\right)^\top \boldsymbol{W}\,\boldsymbol{\sigma}(\boldsymbol{y})$. Neuron $i$ dies when sigmoid saturates[2] for some $y_i \gg 1$. After that for all $\widetilde{\boldsymbol{y}} = \boldsymbol{y} + \alpha\boldsymbol{e}_i$, with $\alpha \geq 0$ the energy has flat region, i.e., $E(\boldsymbol{y}) = E(\widetilde{\boldsymbol{y}}(\alpha))$ for all admissible $\alpha$.*

**Example 4** (flat energy with softmax activations). *Activation $softmax(\boldsymbol{y}) = \exp(\boldsymbol{y})\big/\sum_{i=1}^N \exp(y_i)$ corresponds to $L(\boldsymbol{y}) = \log\left(\sum_{i=1}^N \exp(y_i)\right)$ and energy $E(\boldsymbol{y}) = (\boldsymbol{y} - \boldsymbol{b})^\top softmax(\boldsymbol{y}) - \log\left(\sum_{i=1}^N \exp(y_i)\right) - \frac{1}{2}\left(softmax(\boldsymbol{y})\right)^\top \boldsymbol{W}\,softmax(\boldsymbol{y})$. Consider variable $\widetilde{\boldsymbol{y}}(\boldsymbol{c}) = \boldsymbol{y} + \boldsymbol{c}$, where $\boldsymbol{c}$ is a vector with all components equal $c \in \mathbb{R}$. So constant shift, $softmax(\boldsymbol{y}) = softmax(\widetilde{\boldsymbol{y}}(\boldsymbol{c}))$, Lagrange function shifts on $c$ and $\widetilde{\boldsymbol{y}}(\boldsymbol{c})^\top softmax(\widetilde{\boldsymbol{y}}(\boldsymbol{c})) = \boldsymbol{y}^\top softmax(\boldsymbol{y}) + c$, so overall energy remains the same, i.e., $E(\boldsymbol{y}) = E(\widetilde{\boldsymbol{y}}(c))$ for arbitrary c.*

The examples above demonstrate that energy function (2) fails to distinguish states on a large fraction of state space. In Examples 2 and 3 this happens because the activation function saturates for some inputs, and in Example 4 this is a consequence of invariance. The latter case is typically not considered as a dead neuron, but since the number of degrees of freedom decreases by one we call this "effective neuron" dead all the same.[3] The formal definition of dead neurons appears below (see Maas et al. (2013), Lu et al. (2019), Cui & Fearn (2018) for the discussion of dead neurons in the context of ReLU activation function).

**Definition** (dead neurons). *At a given point $\boldsymbol{y} \in \mathbb{R}^N$ activation function $\boldsymbol{g}$ has $k$ dead neurons if it is possible to find $\boldsymbol{V} \in \mathbb{R}^{N \times k}$ such that $\boldsymbol{g}\left(\boldsymbol{y} + \boldsymbol{V}\boldsymbol{c}\right) = \boldsymbol{g}\left(\boldsymbol{y}\right)$ for all $\boldsymbol{c} \in \mathbb{R}_+^k$.*

The examples above hint that most of the activation functions will lead to dead neurons. The first large class is activation functions with saturation: hyperbolic tangent, sigmoid, ReLU, GELU, SiLU,

---

[2]Formally, the sigmoid function saturates in the limit $y \to \pm\infty$. However, for computations in floating-point arithmetic it is safe to say that sigmoid saturates when the absolute value of the argument is sufficiently large but finite.

[3]One can also say that softmax is always saturated. If we consider input $\boldsymbol{y}$ in the new basis where $\widetilde{y}_1 = \sum_i y_i$, it is easy to see that after the softmax this component becomes 1 regardless of the input.

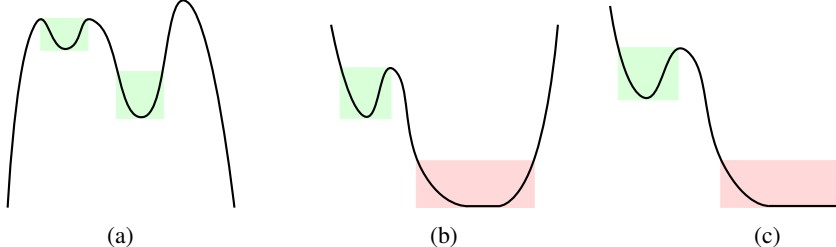

Figure 2: Sketch of three problematic energy functions: (a) unbounded from below with two stable states (we analyze this situation in Appendix C), (b) bounded from below but with compact flat region, (c) bounded from below but with non-compact flat region. According to Proposition 1 case (c) is realized in models Krotov & Hopfield (2020) when neurons die. In red regions stability properties do not follow from Lyapunov theorems. LaSalle's invariance principle (Haddad & Chellaboina, 2008, Theorem 3.3) ensures that in case (b) isolated steady states are stable, but for the case (c) separate stability analysis is needed (see Section 3.2 for details).

SELU, Gaussian, etc.[4] The second class is activation functions with symmetries, e.g., softmax and layer norm. With the definition of dead neurons, we can formalize a pathological behavior of energy function (2).

**Proposition 1.** *If at a given point $\boldsymbol{y}$ activation function $\boldsymbol{g}$ has $k$ dead neuron defined by $\boldsymbol{V} \in \mathbb{R}^{N \times k}$, energy function (2) has constant value in a subspace $\mathcal{D} = \left\{ \boldsymbol{y} + \boldsymbol{V}\boldsymbol{c} : \boldsymbol{c} \in \mathbb{R}_+^k \right\}$.*

*Proof: Appendix B.*

Flat energy directions are illustrated in Figure 1 and Figure 2c. Intuitively, it is clear that having large regions with flat energy is not good. In the next section, we explain in detail what will go awry.

As illustrated in Figure 2 energy function can appear problematic for reasons not directly related to dead neurons. We argue that these two situations are less grim. Energy illustrated in Figure 2a can still be utilized with proper initial conditions, e.g., $\|\boldsymbol{y}_0\|_2 \leq R$ for some small $R$. Besides, we demonstrate in Appendix C that restricting parameters to ensure energy is bounded from below can lead to catastrophic capacity reduction. Energy from Figure 2b is less problematic for stability analysis because invariance principle (Haddad & Chellaboina, 2008, Theorem 3.3) can be used.

## 3 CONSEQUENCES OF DEAD NEURONS

Proposition 1 implies that for most architectures used in practice, there are large regions of state space with flat energy functions. This has several negative consequences that we somewhat arbitrarily classify as problems with sensitivity and stability.

### 3.1 SENSITIVITY

As evident from Figure 1 energy is flat in directions $\boldsymbol{V}$ corresponding to dead neurons. This can be formalised if one considers new variables $\boldsymbol{y}_d = \boldsymbol{V}\boldsymbol{V}^\top \boldsymbol{y}$, $\boldsymbol{y}_a = \left(\boldsymbol{I} - \boldsymbol{V}\boldsymbol{V}^\top\right)\boldsymbol{y}$ corresponding to dead and alive neurons. Since $\boldsymbol{y} = \boldsymbol{y}_d + \boldsymbol{y}_a$, the energy (2) becomes

$$E(\boldsymbol{y}_d, \boldsymbol{y}_a) = (\boldsymbol{y}_d + \boldsymbol{y}_a - \boldsymbol{b})^\top \boldsymbol{g}(\boldsymbol{y}_a) - L(\boldsymbol{y}_d + \boldsymbol{y}_a) - \frac{1}{2} \left(\boldsymbol{g}(\boldsymbol{y}_a)\right)^\top \boldsymbol{W}\boldsymbol{g}(\boldsymbol{y}_a)$$

$$= (\boldsymbol{y}_a - \boldsymbol{b})^\top \boldsymbol{g}(\boldsymbol{y}_a) - L(\boldsymbol{y}_a) - \frac{1}{2} \left(\boldsymbol{g}(\boldsymbol{y}_a)\right)^\top \boldsymbol{W}\boldsymbol{g}(\boldsymbol{y}_a),$$

where we used $\boldsymbol{g}(\boldsymbol{y}_d + \boldsymbol{y}_a) = \boldsymbol{g}(\boldsymbol{y}_a)$ and $L(\boldsymbol{y}_d + \boldsymbol{y}_a) = L(\boldsymbol{y}_a) + \boldsymbol{g}(\boldsymbol{y}_a)^\top \boldsymbol{y}_d$.

---

[4]For sigmoid, hyperbolic tangent and Gaussian function the saturation is precise only in the limit $x \to \pm\infty$. However, as explained in the Example 3 in the floating-point arithmetics these functions effectively saturate for finite values of the argument. Moreover, even with arbitrary precision, approximately flat direction can not be used to reliably store any information since it will require weights with exponentially large magnitudes.

We see that energy does not depend on variables $\boldsymbol{y}_d$ corresponding to dead neurons, which means the effective number of degrees of freedom is decreased from $N$ to $N - k$ where $k$ is the number of dead neurons. If only energy is available it is not possible to recover values on $\boldsymbol{y}_d$ and steady states can not be found as in examples from Figure 1.

Variables $\boldsymbol{y}_d$ are from the kernel of $\boldsymbol{g}$ which means the activation function becomes effectively invertible. In Hopfield (1984), the energy function for this situation has already been introduced. It is instructive to compare a new energy function Krotov & Hopfield (2020) with the old one. Lyapunov function from the 1984 paper, in the notation used in this article, reads

$$\widetilde{E}(\boldsymbol{u}) = \sum_i \int_0^{g_i(u_i)} d\tau_i g_i^{-1}(\tau_i) - \boldsymbol{b}^\top \boldsymbol{g}(\boldsymbol{u}) - \frac{1}{2} \left(\boldsymbol{g}(\boldsymbol{u})\right)^\top \boldsymbol{W} \boldsymbol{g}(\boldsymbol{u}), \tag{3}$$

and dynamical system is precisely the same as (1). Activation functions $g_i$ are invertible and monotone by assumption. In Appendix D we explain precisely how to adapt the model from Hopfield (1984) to our context. The first term of (3) is seemingly distinct from any term of (2). To show that this is an illusion we simplify integrals as follows

$$\int_0^{g(u)} d\tau g^{-1}(\tau) = \int_{g^{-1}(0)}^u d\rho\, g'(\rho)\rho = - \int_{g^{-1}(0)}^u d\rho\, g(\rho) + \rho g(\rho)\big|_{g^{-1}(0)}^u = - \int_{g^{-1}(0)}^u d\rho\, g(\rho) + u g(u),$$

where in the first step we used $\tau = g(\rho)$. Using this simplification and additional definition of Lagrange function $L(\boldsymbol{u}) : \boldsymbol{g}(\boldsymbol{u}) = \frac{\partial L(\boldsymbol{u})}{\partial \boldsymbol{u}}$ we immediately recognize that energy function from Krotov & Hopfield (2020) is precisely the same as from Hopfield (1984) but without the assumption that $\boldsymbol{g}$ should be invertible. New energy from Krotov & Hopfield (2020) does not formally contain the inverse of the activation function, but as we see still implies that $\boldsymbol{g}$ is invertible.

Besides insensitivity to values of dead neurons, there is another problematic fact. Proposition 1 implies that, when at least one neuron is dead, the energy function has invariant transformation $E(\boldsymbol{y}) = E(\boldsymbol{y} + \boldsymbol{V}\boldsymbol{c})$. The steady state of dynamical system (1) does not have this symmetry, so, when one transforms $\boldsymbol{y} \to \boldsymbol{y} + \boldsymbol{V}\boldsymbol{c}$ and preserve energy, this corresponds to a new dynamical system with $\boldsymbol{b} \to \boldsymbol{b} - \boldsymbol{V}\boldsymbol{c}$. In other words, in the regions when dead neurons are present, energy is not sensitive to the changes in the bias term.

We summarise arguments made in this section in the proposition below

**Proposition 2.** *For energy function (2) the following is true:*

1. *Energy function does not depend on variables from the kernel of $\boldsymbol{g}$, so one can always assume that activation function $\boldsymbol{g}$ is invertible.*

2. *Suppose at a given point $\boldsymbol{y}$ dead neurons are described by matrix $\boldsymbol{V} \in \mathbb{R}^{N \times k}$. In this case energy function $E(\boldsymbol{y})$ is Lyapunov function for dynamical systems (1) with $\boldsymbol{b} = \widetilde{\boldsymbol{b}} - \boldsymbol{V}\boldsymbol{c}$ for any $\boldsymbol{c} \in \mathbb{R}_+^k$.*

### 3.2 STABILITY OF DYNAMICAL SYSTEM

One of the main applications of the Lyapunov function is to perform stability analysis Haddad & Chellaboina (2008). This part is relevant for predictive coding and related approaches Xie & Seung (2003), Millidge et al. (2022b) to ensure the existence of at least some steady states. Besides, when associative memory is considered, one can only recover stable steady states, so stability directly influences memory capacity. Lyapunov function also characterizes a basin of attraction for a given steady state. In the context of associative memory, basins of attraction define a measure of similarity between inputs, since inputs from the same basin result in the recovery of the same memory. Below, we will show that in the regions with dead neurons, Lyapunov function (2) alone fails to help in stability analysis.

Stability analysis with the Lyapunov function boils down to the study of the energy landscape near the steady state. Using the Taylor series, we find

$$E(\boldsymbol{y}^\star + \boldsymbol{\delta}) - E(\boldsymbol{y}^\star) = \boldsymbol{\delta}^\top \frac{\partial E(\boldsymbol{u})}{\partial \boldsymbol{u}}\bigg|_{\boldsymbol{u}=\boldsymbol{y}^\star} + \frac{1}{2}\boldsymbol{\delta}^\top \frac{\partial^2 E(\boldsymbol{u})}{\partial \boldsymbol{u}^2}\bigg|_{\boldsymbol{u}=\boldsymbol{y}^\star + t\boldsymbol{\delta}} \boldsymbol{\delta}, \ t(\boldsymbol{\delta}, \boldsymbol{y}^\star) \in (0, 1),$$

where $y^\star : Wg(y^\star) - y^\star + b = 0$. Computing derivatives, we find $E(y^\star + \delta) - E(y^\star) = \delta^\top (\Lambda(y^\star) - \Lambda(y^\star)W\Lambda(y^\star)) \delta, \|\delta\|_2 \ll 1$. For stability in the vicinity of a steady state, it is sufficient that the matrix above is positive definite, and for instability, it is sufficient that at least one eigenvalue of this matrix is negative. In situations such as in Figure 1 and Figure 2c when energy has a flat direction nothing can be said about the dynamics in this region[5]. It is easy to show that if dead neurons are present at point $u$, matrix $V$ lays in zero eigenspace of $\Lambda(u)$. To show this we consider the Taylor series $g(u + Vc) = g(u) + \Lambda(u + tVc)Vc, t \in (0, 1)$, and since $g(u + Vc) = g(u)$ we confirm that $\Lambda V = 0$, i.e., zero eigenspace is always present.

Since energy is insufficient to analyze stability, we need to use a dynamical system (1) directly. We suppose that in the region of interest $V$, matrix that describes dead neurons, does not change and use $y_d, y_d$ defined in the previous section. With that, one can show that dynamical system becomes

$$\dot{y}_a(t) = P_a Wg(y_a(t)) - y_a + P_a b, \ \dot{y}_d(t) = P_d Wg(y_a(t)) - y_d + P_d b, \quad (4)$$

where $P_d = VV^\top$ and $P_a = I - P_d$.[6] One can immediately see that dead neurons do not influence stability and, when steady state $y_a^\star$ is reached, one can find $y_d^\star = P_d Wg(y_a^\star) + P_d b$. Moreover, when $y_a(t)$ reaches steady state $y_d(t)$ converges exponentially from the arbitrary starting point, i.e., the whole flat region is a basin of attraction.

The sufficient condition for stability/instability of $y_a$ can be obtained from (4) with linearisation. More specifically, stability is defined by matrix $S = V_\perp^\top W V_\perp V_\perp^\top \Lambda V_\perp - I \equiv W_\perp \Lambda_\perp - I \in \mathbb{R}^{(N-k)\times(N-k)}$, where $V_\perp \in \mathbb{R}^{N \times N-k}$ is a matrix with orthogonal columns such that $P_a = V_\perp V_\perp^\top$. When matrix $S$ does not have eigenvalues with a positive real part, the steady state is stable; in case there is at least one eigenvalue with a positive real part, the steady state is unstable (see (Haddad & Chellaboina, 2008, Theorem 3.19)).

The structure of matrix $S$ suggests that, in fact, dynamics is stable when energy function indicates that. Indeed, $\Lambda_\perp^{1/2} W_\perp \Lambda_\perp^{1/2}$ has the same spectrum as $W_\perp \Lambda_\perp$, so sufficient condition for $S$ is equivalent to sufficient condition for $\Lambda_\perp^{1/2} W_\perp \Lambda_\perp^{1/2} - I$. Finally since $\Lambda_\perp$ is full rank we find that if $\Lambda_\perp W_\perp \Lambda_\perp - \Lambda_\perp$ is negatively stable (has non-positive eigenvalues), the dynamics of $y_a$ is stable. Since this matrix is a restriction of $\Lambda W \Lambda - \Lambda$ on the range of Hessian we conclude that one can analyze the stability of steady-state using energy if projector on the range of Hessian is known. It is important to note that matrix $\Lambda W \Lambda - \Lambda$ can have other flat directions not corresponding to nullspace of Hessian, these directions will cause instability, so one need to distinguish them from directions corresponding to dead neurons.

Clearly, zero modes of $\Lambda$ do not negatively affect the spectrum of matrix $S$. On the contrary, one may expect a stabilizing effect since Cauchy interlacing theorem (Horn & Johnson, 2012, Theorem 4.3.28, Corollary 4.3.37) ensures that the spectrum of $W_\perp$ is in-between minimal and maximal eigenvalues of $W$. So when $\lambda(W\Lambda) \geq 1$ it is likely that $\lambda(W_\perp \Lambda_\perp) < 1$. To demonstrate the effect, one needs to make some assumptions about the spectrum of $W$. Given a standard deep learning practice to use random matrices for initialization of model weights Glorot & Bengio (2010), He et al. (2015), it is natural to assume that $W$ is drawn from Gaussian orthogonal ensemble. Under this condition, one may show a stabilization effect, as demonstrated next.

**Proposition 3.** *Suppose $W \in \mathbb{R}^{N\times N}$ is a matrix from a Gaussian orthogonal ensemble. If $k$ neurons are dead $\mathbb{E}\|W_\perp\|_2 = \sqrt{1 - \frac{k}{N}}\mathbb{E}\|W\|_2$ holds for large $N$.*

*Proof: Appendix F.*

We gather the main results of this section in the proposition that follows.

**Proposition 4.** *For energy function (2) the following is true:*

1. *Flat directions of dead neurons characterised by $V$ form nullspace of Hessian $\Lambda$.*

---

[5]Note, however, that for the face given in Figure 2b it is possible to construct a positively invariant set based on energy levels. After that LaSalle's invariance principle (Haddad & Chellaboina, 2008, Theorem 3.3) guarantees that isolated steady states are asymptotically stable. This material is standard and covered in many books, e.g., (Haddad & Chellaboina, 2008, Section 3.3.).

[6]We assume for simplicity that $V$ has orthonormal columns.

2. *Steady state is stable if energy function predicts local stability in the range of Hessian, i.e. if matrix $\boldsymbol{\Lambda}\boldsymbol{W}\boldsymbol{\Lambda} - \boldsymbol{\Lambda}$ restricted on the range of Hessian is positive definite.*

3. *If one performed direct optimization of the energy function and found that $\widetilde{\boldsymbol{y}}$ corresponds to the minimum, it is possible to recover the steady state of dynamical system (1) as follows: (i) compute the projector $\boldsymbol{P}_d$ on the nullspace of $\boldsymbol{\Lambda}$; (ii) steady state of dynamical system reads $\boldsymbol{y} = \widetilde{\boldsymbol{y}} + \boldsymbol{P}_d\left(\boldsymbol{W}\boldsymbol{g}(\widetilde{\boldsymbol{y}}) + \boldsymbol{b}\right)$.*

4. *When steady state $(\boldsymbol{y}_a^\star, \boldsymbol{y}_d^\star)$ is stable, the whole non-compact flat region $(\boldsymbol{y}_a^\star, \boldsymbol{y}_d^\star + \boldsymbol{V}\boldsymbol{c})$ is basin of attraction.*

## 4 ASSOCIATIVE MEMORIES WITHOUT DEAD NEURONS

We have seen that dead neurons present certain problems for energy function (2): (i) flat regions make optimization and analysis of energy cumbersome; (ii) one needs to distinguish between harmless flat direction of Hessian and other flat direction of energy that can compromise stability; (iii) energy do not contain full information and one need to build a projector on the nullspace of Hessian to restore the whole state. The last point implies that one needs Hessian even when stability analysis is not performed and one merely wants to compute a local minimum of energy which can often be done successfully with first-order methods.

To avoid these difficulties we will slightly modify the dynamics of associative memory (1) to minimally alter steady states and get a better energy function. In a simplified terms, our idea is to modify right-hand side of (1) such that it becomes conservative vector field. If this is the case, flat energy directions disappear. For example, the simplest model of this kind is $\dot{\boldsymbol{u}}(t) = \boldsymbol{W}^\top\boldsymbol{g}(\boldsymbol{W}\boldsymbol{u}(t) + \boldsymbol{b}) - \boldsymbol{u}(t) = -\frac{\partial E(\boldsymbol{u})}{\partial \boldsymbol{u}}$ with energy $E(\boldsymbol{u}) = \frac{1}{2}\boldsymbol{u}^\top\boldsymbol{u} - L(\boldsymbol{W}\boldsymbol{u} + \boldsymbol{b})$. A more general dynamical system along the same lines reads:

$$\dot{\boldsymbol{u}}(t) = \boldsymbol{R}(\boldsymbol{u})\left(\boldsymbol{g}(\boldsymbol{W}\boldsymbol{u}(t) + \boldsymbol{b}) - \boldsymbol{u}(t)\right), \ \boldsymbol{u}(0) = \boldsymbol{u}_0, \tag{5}$$

where $\boldsymbol{R}(\boldsymbol{u})$ is a matrix-valued function to be specified later.

If one assumes that it is possible to select $\boldsymbol{R}(\boldsymbol{u})$ that has empty nullspace for all $\boldsymbol{u}$, i.e., that information is not lost after multiplication by $\boldsymbol{R}(\boldsymbol{u})$, the condition that we will verify later, steady states of the newly introduced system (5) follows from the old one with affine transformation $\boldsymbol{y} = \boldsymbol{W}\boldsymbol{u} + \boldsymbol{b}$. This means all architectures possible with the old model (1) are also possible with the new one (5). More precisely, the structure of steady state is the same for memory (1) and memory (5), but the structure of basins of attraction is different owing to the matrix $\boldsymbol{R}(\boldsymbol{u})$ absent in model (1).

Dynamical systems (5) and (1) have equivalent steady states, but (5) allows to construct of a large set of Lyapunov functions with good properties in a systematic manner.

**Proposition 5.** *For a parametric energy function*

$$E(\boldsymbol{u}; \alpha, \beta, \gamma, \boldsymbol{S}) = \alpha E_1(\boldsymbol{u}) + \beta E_2(\boldsymbol{u}) + \gamma E_3(\boldsymbol{u}; \boldsymbol{S}), \alpha, \beta, \gamma \in \mathbb{R},$$

$$E_1(\boldsymbol{u}) = \boldsymbol{u}^\top\boldsymbol{W}\boldsymbol{g}\left(\boldsymbol{W}\boldsymbol{u} + \boldsymbol{b}\right) - L\left(\boldsymbol{W}\boldsymbol{u} + \boldsymbol{b}\right) - \frac{1}{2}\boldsymbol{g}\left(\boldsymbol{W}\boldsymbol{u} + \boldsymbol{b}\right)^\top\boldsymbol{W}\boldsymbol{g}\left(\boldsymbol{W}\boldsymbol{u} + \boldsymbol{b}\right),$$

$$E_2(\boldsymbol{u}) = \frac{1}{2}\boldsymbol{u}^\top\boldsymbol{W}\boldsymbol{u} - L\left(\boldsymbol{W}\boldsymbol{u} + \boldsymbol{b}\right), \ E_3\left(\boldsymbol{u}; \boldsymbol{S}\right) = \frac{1}{2}\left(\boldsymbol{u} - \boldsymbol{g}(\boldsymbol{W}\boldsymbol{u} + \boldsymbol{b})\right)^\top\boldsymbol{S}\left(\boldsymbol{u} - \boldsymbol{g}(\boldsymbol{W}\boldsymbol{u} + \boldsymbol{b})\right) \tag{6}$$

*the following is true:*

1. *$E(\boldsymbol{u}; \alpha, \beta, \gamma, \boldsymbol{S})$ is a Lyapunov function for dynamical system (5) if one can find $\boldsymbol{Q} \geq 0$, $\boldsymbol{R}(\boldsymbol{u})$, $\boldsymbol{S} = \boldsymbol{S}^\top$ such that for all $\boldsymbol{u}$*

   $$\boldsymbol{R}^\top\boldsymbol{F}(\boldsymbol{u}) + \boldsymbol{F}(\boldsymbol{u})^\top\boldsymbol{R} \geq \boldsymbol{Q}, \boldsymbol{F}(\boldsymbol{u}) = \boldsymbol{W}\boldsymbol{\Lambda}\left(\boldsymbol{W}\boldsymbol{u} + \boldsymbol{b}\right)\left(\alpha\boldsymbol{W} - \gamma\boldsymbol{S}\right) + \beta\boldsymbol{W} + \gamma\boldsymbol{S}. \tag{7}$$

2. *For arbitrary parameters $\alpha, \beta, \gamma, \boldsymbol{S}$ one can always find orthogonal matrix $\boldsymbol{R}(\boldsymbol{u})$ such that $E(\boldsymbol{u}; \alpha, \beta, \gamma, \boldsymbol{S})$ is a Lyapunov function for dynamical system (5).*

3. *$E_2(\boldsymbol{u})$ and $E_3(\boldsymbol{u}; \boldsymbol{S})$ does not have flat directions corresponding to dead neurons.*

4. *Sufficient condition for stability of steady-state* $\boldsymbol{u}^\star : \boldsymbol{u}^\star = \boldsymbol{g}\left(\boldsymbol{W}\boldsymbol{u}^\star + \boldsymbol{b}\right)$ *is*

$$\boldsymbol{F}(\boldsymbol{u}^\star)\left(\boldsymbol{I} - \boldsymbol{\Lambda}\left(\boldsymbol{W}\boldsymbol{u}^\star + \boldsymbol{b}\right)\boldsymbol{W}\right) + \left(\boldsymbol{I} - \boldsymbol{W}\boldsymbol{\Lambda}\left(\boldsymbol{W}\boldsymbol{u}^\star + \boldsymbol{b}\right)\right)\left(\boldsymbol{F}(\boldsymbol{u}^\star)\right)^\top > 0 \qquad (8)$$

*Proof: Appendix E.*

Proposition 5 allows one to make several useful conclusions.

First, the construction of the Lyapunov function boils down to the analysis of matrix inequality (7). In general, this inequality is nonlinear, but in particular cases, it reduces to a Sylvester equation which can be systematically analyzed Simoncini (2016). Below we will provide several explicit choices of $\boldsymbol{R}$ that result in a valid Lyapunov function.

The second point of Proposition 5 shows that suitable $\boldsymbol{R}$ always exist. Moreover, this $\boldsymbol{R}$ does not alter the steady state, since it is an orthogonal matrix. Whether this choice of $\boldsymbol{R}$ is practical depends on the parametrization of weights and other details of the model.

The third point shows that unless $\beta = \gamma = 0$, the energy function does not have a flat direction, and the fourth point gives a sufficient condition for the stability of a particular state. This part is harder to analyze in general, so we will discuss this condition for selected examples below.

Finally, Proposition 5 does not require Hessian to be positive definite which allows one to use a richer set of activation functions.

**Example 5** ($\alpha = \gamma = 0$, $\beta = 1$). *In this case $E = E_2$ condition (7) becomes $\boldsymbol{R}^\top \boldsymbol{W} + \boldsymbol{W}\boldsymbol{R} \geq 0$. There are many strategies to select $\boldsymbol{R}$ and $\boldsymbol{W}$:*

1. *The simplest choice is to take $\boldsymbol{R} = \boldsymbol{W}$ since in this case condition (7) reduces to $\boldsymbol{W}^2 \geq 0$ which is true for arbitrary symmetric matrix.*

2. *Unfortunately, if matrix $\boldsymbol{W}$ has low rank and one takes $\boldsymbol{R} = \boldsymbol{W}$, dynamical system (5) can lead to false memories whenever $\boldsymbol{f}(\boldsymbol{u}(t))$ reaches a nullspace of $\boldsymbol{W}$. To exclude this possibility one can parametrize $\boldsymbol{W} = \boldsymbol{O}\boldsymbol{P}$ in terms of its polar decomposition and select $\boldsymbol{R} = \boldsymbol{O}$. With that choice condition (7) becomes $\boldsymbol{P} \geq 0$ which automatically follows from polar decomposition.*

3. *We can restrict $\boldsymbol{W}$ to be positive definite by taking $\boldsymbol{W} = \boldsymbol{K}\boldsymbol{K}^\top$ for some full-rank $\boldsymbol{K}$. In this case, a large set of positive definite matrices $\boldsymbol{R}$ exists such that $\boldsymbol{R}\boldsymbol{W} + \boldsymbol{W}\boldsymbol{R} \geq 0$. More specifically, one needs to restrict the condition number of $\boldsymbol{R}$ as explained in Nicholson (1979).*

4. *If $\boldsymbol{W}$ is positive definite the other option is to select arbitrary $\boldsymbol{Q} > 0$ and consider Lyapunov equation $\boldsymbol{R}\boldsymbol{W} + \boldsymbol{W}\boldsymbol{R} = \boldsymbol{Q}$ which is known to have unique solution for arbitrary $\boldsymbol{Q} > 0$. One can explicitly provide it in many forms Lancaster (1970), Simoncini (2016), for example $\boldsymbol{R} = \int_0^\infty d\tau \exp(-\tau\boldsymbol{W})\boldsymbol{Q}\exp(-\tau\boldsymbol{W})$.*

*Condition for the stability of state (8) in this case reduces to $\boldsymbol{W} - \boldsymbol{W}\boldsymbol{\Lambda}\left(\boldsymbol{W}\boldsymbol{u}^\star + \boldsymbol{b}\right)\boldsymbol{W} > 0$. Note that nullspace of $\boldsymbol{\Lambda}$ does not compromise stability since it does not transfer to nullspace of $\boldsymbol{W} - \boldsymbol{W}\boldsymbol{\Lambda}\boldsymbol{W}$.*

Interestingly, one can also construct associative memory with non-symmetric weights as explained in the next two examples.

**Example 6** ($\alpha = \beta = 0$, $\gamma = 1$ and $\boldsymbol{W} \neq \boldsymbol{W}^\top$). *$E_3$ is the only energy function that is defined for $\boldsymbol{W} \neq \boldsymbol{W}^\top$. If we account for that in Proposition 5, existence condition (7) becomes $\boldsymbol{R}^\top\left(\boldsymbol{I} - \boldsymbol{W}^\top\boldsymbol{\Lambda}\right)\boldsymbol{S} + \boldsymbol{S}\left(\boldsymbol{I} - \boldsymbol{\Lambda}\boldsymbol{W}\right)\boldsymbol{R} \geq 0$. Two examples of suitable $\boldsymbol{R}, \boldsymbol{S}, \boldsymbol{W}$ are given below:*

1. *The simplest choice is to take $\boldsymbol{S} = \boldsymbol{I}$ and $\boldsymbol{R} = \boldsymbol{I} - \boldsymbol{W}^\top\boldsymbol{\Lambda}$. One also may ensure that $\boldsymbol{R}$ is invertible with $\|\boldsymbol{W}\|_2 < \|\boldsymbol{\Lambda}\|_2^{-1}$.*

2. *Another possible choice is to take $\boldsymbol{R} = \boldsymbol{\Lambda}$, and ensure $\boldsymbol{\Lambda} - \frac{1}{2}\boldsymbol{\Lambda}\left(\boldsymbol{W} + \boldsymbol{W}^\top\right)\boldsymbol{\Lambda} \geq 0$, which can be simplified to $\omega(\boldsymbol{W}) \leq \lambda_{\min}(\boldsymbol{\Lambda})/\lambda_{\max}(\boldsymbol{\Lambda}^2)$ where $\omega(\boldsymbol{W})$ is numerical abscissa $\omega(\boldsymbol{W}) = \lambda_{\max}(\boldsymbol{W} + \boldsymbol{W}^\top)/2$. For example, if $\boldsymbol{\Lambda} \geq 0$ any $\boldsymbol{W}$ with non-positive real part of the spectrum suffices.*

**Example 7** (associative memory with smooth Leaky ReLU). *Here we build a concrete example of hierarchical associative memory Krotov (2021) with smooth Leaky ReLU Biswas et al. (2021) to avoid dealing with ODE having a non-smooth right-hand side. The activation function reads $g(u) = u \left(5 + 3\mathrm{erf}\left(3u/8\right)\right)/8$ and its derivative is $g^{'}(u) = g(u)/u + 9u \exp\left(-9u^2/64\right)/\left(32\sqrt{\pi}\right)$, where all operations are pointwise and $\mathrm{erf}$ is error function. Weights and state vector are selected as in Example 1 but without requirement $W_{i,i+1} = W_{i+1,i}^{\top}$. After that we select $R = I - W^{\top}\Lambda$ and obtain the following dynamical system*

$$\dot{u}(t) = \left(I - W^{\top} \odot g^{'}(Wu(t) + b)\right)(g(Wu(t) + b) - u(t)), \; u(0) = u_0.$$

*For this system Lyapunov function $E_2 = \frac{1}{2}\left(g(Wu(t) + b) - u(t)\right)^{\top}\left(g(Wu(t) + b) - u(t)\right)$ is non-increasing on trajectories. Matrix $\left(I - W^{\top} \odot g^{'}(Wu(t) + b)\right)$ can in principle have non-trivial nullspace. To avoid this one observes that $\|W\|_2 \leq \max_i \|W_{i,i+1}\|_2 + \max_i \|W_{i,i-1}\|_2$ (the bound is tight) and take $W_{i,i+1} = \widetilde{W}_{i,i\pm 1}\Big/\left(2\left\|\widetilde{W}_{i,i\pm 1}\right\|_2\right)$, for some $\widetilde{W}$ with the same block structure. Note, that in this construction $\Lambda$ is not positive definite, and $W$ is not symmetric.*

All theoretical statements that we derived are valid only for modified dynamical system 5. However, it is easy to see that energy function $E_3$ from Proposition 5 can be adopted for the original temporal dynamics (1). The resulting energy function and dynamical system reads

$$\dot{y}(t) = R(y(t))\left(Wg(y(t)) - y(t) + b\right), \; y(0) = y_0, \tag{9}$$

$$E_3(y) = \frac{1}{2}\left(Wg(y(t)) - y(t) + b\right)^{\top}S\left(Wg(y(t)) - y(t) + b\right). \tag{10}$$

We used temporal dynamics (9) with $R = I - W\Lambda(y)$ and energy function (10) with $S = I$ to generate results given in Figure 1. One can also prove results similar to Proposition 5 for energy function (10) but since the extension is straightforward we do not pursue this further.

## 5 CONCLUSIONS

We describe the effect of dead neurons on the energy landscape of associative memory, analyze the consequences for stability, and provide several remedies, including new dynamical systems with good energy functions. We think it is appropriate to discuss the overall significance of the Lyapunov function for associative memory. Is it necessary to have this function at all? How is this function used in practice?

One may observe that, currently, the Lyapunov function is underutilized. As a rule, one uses several steps or even a single step of temporal dynamics of ODE during training and inference Hoover et al. (2024), Hoover et al. (2022), Millidge et al. (2022a), Ramsauer et al. (2020), Krotov & Hopfield (2016). The role of Lyapunov's function is merely to provide comfort that some steady states may exist somewhere. As we show in this article, it is often a false comfort.

It is important to note, that the Lyapunov function in itself does not ensure stability: (i) it may be the case that no steady state exists, (ii) limit cycles may still be present, (iii) all steady states may be unstable. Besides that, for any steady state, the Lyapunov function can be constructed (by solving the Lyapunov equation) even when the "global" Lyapunov function is unavailable.

The Lyapunov function, as it is currently considered, is not of huge help. We can suggest several appropriate use cases: (i) model parameters may be adjusted on the training stage to make the Lyapunov function unstable for particular states – something that can not be easily done by integrating the dynamical system alone. This may help to learn from negative and adversarial examples Wang et al. (2024), Goodfellow et al. (2014); (ii) Lyapunov function may help in theoretical understanding of memory capacity, e.g., in the present article we have already shown that a whole flat direction corresponds to the basin of attraction, and can not support more than a single memory; (iii) Lyapunov function may be directly used to compute and manipulate basins of attraction, potentially speeding up learning and making memory more robust to adversarial attacks.

All in all, we think that the Lyapunov function is a powerful tool that can lead to novel theoretical results and practical learning techniques in the field of associative memory. We hope that our results will inspire further research in this direction.

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

## A  RELATIONS BETWEEN MEMORY MODELS

Our model is most directly related to Dense Associative Memory described in (Krotov, 2021, Equation (2)). This equation reads

$$\tau_I \frac{dx_I}{dt} = \sum_{J=1}^{N} W_{IJ} g_J - x_I,$$

where $x_I, I = 1, \ldots, N$ are activities of individual neurons, $g_J = \frac{dL(\{X_I\})}{dx_J}$ is activation function which is a gradient of Lagrange function $L$ that depends on all activities, and $\tau_I$ control relaxation

time of individual neurons. Energy function used in this work is available in equation (Krotov, 2021, Equation (3)) and is given below:

$$E = \sum_{I=1}^{N} x_I g_I - L - \frac{1}{2} \sum_{I,J=1}^{N} g_I W_{IJ} g_J.$$

It is easy to see that there is a direct relation between dynamical system (1) and the model (2) we use in the article and the ones from Krotov (2021): $W_{IJ}$ are the elements of matrix $\boldsymbol{W}$, activities $x_I$ are components of $\boldsymbol{y}$. Two differences between the model from that article and Dense Associative Memory are: bias term that we use $\boldsymbol{b}$ is absent from Dense Associative Memory Krotov (2021), time scales $\tau_I$ are absent from our model. The bias term can be simply removed by taking $\boldsymbol{b} = 0$ and since it is not used in the stability analysis, this does not affect our results. Observe, that if $\boldsymbol{b} = 0$ the energy that we use (2) is precisely the same as in Krotov (2021). Similarly, time scale $\tau_I$ are absent from the definition of energy function in Krotov (2021) so they do not influence flat energy regions. However, they mildly influence stability analysis since after linearisation Jacobian will by multiplied by $\boldsymbol{D}^{-1}$ where $\boldsymbol{D}$ is a diagonal matrix containing time scales on the diagonal.

Next we discuss the relation to model (Krotov & Hopfield, 2020, Equation (1)). This model seems to be quite different from our formulation, but it is a specific case of Dense Associative Memory (Krotov, 2021, Equation (2)) with bias term. To see this we describe model given in (?, Equation (1)) In the article Krotov & Hopfield (2020) authors split neurons on two parts: activations of memory neurons are $h_\mu, \mu = 1, \ldots, N_h$ and activations of feature neurons are $v_i, i = 1, \ldots, N_f$. These neurons has distinct activations function $f_\mu = f(h_\mu)$ and $g_i = g(h_i)$ which are defined as derivatives of corresponding Lagrange functions $f_\mu = \frac{\partial L_h(\{h_\mu\})}{\partial h_\mu}$ and $g_i = \frac{\partial L_v(\{v_i\})}{\partial v_i}$ (Krotov & Hopfield, 2020, Equation (3)). Dynamical equations for these activations (Krotov & Hopfield, 2020, Equation (1)) is reproduced below

$$\tau_f \frac{dv_i}{dt} = \sum_{\mu=1}^{N_h} \xi_{i\mu} f_\mu - v_i + I_i,$$

$$\tau_h \frac{dh_\mu}{dt} = \sum_{i=1}^{N_f} \xi_{\mu i} g_i - h_\mu,$$

where $\xi_{\mu i} = \xi_{i\mu}$ are symmetric weights describing "weights of synapses" and $I_i$ is "the input current into feature neurons." Lyapunov function given in (Krotov & Hopfield, 2020, Equation (2)) reads

$$E = \left[ \sum_{i=1}^{N_f} (v_i - I_i) g_i - L_v \right] + \left[ \sum_{\mu=1}^{N_h} h_\mu f_\mu - L_h \right] - \sum_{\mu,i} f_\mu \xi_{\mu i} g_i.$$

To see that this model is related to (Krotov, 2021, Equation (2)) , consider a special weight matrix $W_{IJ}, x_I, \tau_I$ that have the following block structure

$$W = \begin{pmatrix} 0 & \xi \\ \xi^\top & 0 \end{pmatrix}, \, x = \begin{pmatrix} v \\ h \end{pmatrix}, \tau = \begin{pmatrix} \tau_f e_v \\ \tau_h e_h \end{pmatrix},$$

where $e_v$ and $e_h$ are identity vector of sizes $N_f, N_h$. With this split and Lagrangian $L(\{x_I\}) = L_v(\{v_i\}) + L_h(\{h_\mu\})$ we have

$$\sum_J W_{IJ} g_J - x_I \Rightarrow W \frac{\partial L}{\partial x} = \begin{pmatrix} 0 & \xi \\ \xi^\top & 0 \end{pmatrix} \begin{pmatrix} g(v) \\ f(h) \end{pmatrix} - \begin{pmatrix} v \\ h \end{pmatrix}$$

so we see that we reproduce (Krotov & Hopfield, 2020, Equation (1)) without bias term $I_i$. The same way we reproduce the energy function but without $I_i$.

Given that, both models (Krotov & Hopfield, 2020, Equation (1)), (Krotov, 2021, Equation (2)) are directly related to (1). The only difference is these models contain $\tau$ variable that speed up or slow down temporal dynamics for selected neurons. Importantly, energy functions from models Krotov & Hopfield (2020), Krotov (2021) are equivalent to (2).

Note that other modifications of memory models are available. The main problematic part that lead to flat energy is a Lagrange function introduced in Krotov & Hopfield (2020). So, whenever this

function is used, the energy will have non-compact flat regions corresponding to dead neurons. To give an example, in Millidge et al. (2022a) authors build a modified model but used a non-trivial Lagrange function $L_h$ in Equation (3). If $f(h) = \text{sep}(h)$ can produce dead neurons, the energy from Millidge et al. (2022a) will have a non-compact flat region. One such situation occurs when the softmax function is used as $\text{sep}(h)$ in (Millidge et al., 2022a, Equation (5)).

## B  PROOF OF PROPOSITION 1: DEAD NEURONS LEAD TO NON-COMPACT FLAT ENERGY REGIONS

We assume that we are at point $\boldsymbol{y}$ in the region where neurons are dead, so there is $\boldsymbol{V}$ such that $\boldsymbol{g}(\boldsymbol{y} + \boldsymbol{V}\boldsymbol{c}) = \boldsymbol{g}(\boldsymbol{y})$ for arbitrary $\boldsymbol{c}$ with positive components. We substitute $\boldsymbol{y} + \boldsymbol{V}\boldsymbol{c}$ into the definition of energy to obtain

$$E(\boldsymbol{y} + \boldsymbol{V}\boldsymbol{c}) = (\boldsymbol{y} + \boldsymbol{V}\boldsymbol{c} - \boldsymbol{b})^\top \boldsymbol{g}(\boldsymbol{y} + \boldsymbol{V}\boldsymbol{c}) - L(\boldsymbol{y} + \boldsymbol{V}\boldsymbol{c}) - \frac{1}{2}(\boldsymbol{g}(\boldsymbol{y} + \boldsymbol{V}\boldsymbol{c}))^\top \boldsymbol{W}\boldsymbol{g}(\boldsymbol{y} + \boldsymbol{V}\boldsymbol{c}).$$

First, we use $\boldsymbol{g}(\boldsymbol{y} + \boldsymbol{V}\boldsymbol{c}) = \boldsymbol{g}(\boldsymbol{y})$ to remove shift from activation functions

$$E(\boldsymbol{y} + \boldsymbol{V}\boldsymbol{c}) = (\boldsymbol{y} + \boldsymbol{V}\boldsymbol{c} - \boldsymbol{b})^\top \boldsymbol{g}(\boldsymbol{y}) - L(\boldsymbol{y} + \boldsymbol{V}\boldsymbol{c}) - \frac{1}{2}(\boldsymbol{g}(\boldsymbol{y}))^\top \boldsymbol{W}\boldsymbol{g}(\boldsymbol{y}).$$

Next, we use Taylor series with mean-value reminder to expand Lagrange function

$$L(\boldsymbol{y} + \boldsymbol{V}\boldsymbol{c}) = L(\boldsymbol{y}) + \left(\left.\frac{\partial L(\boldsymbol{z})}{\partial \boldsymbol{z}}\right|_{\boldsymbol{z} = \boldsymbol{y} + \tau\boldsymbol{V}\boldsymbol{c}}\right)^\top \boldsymbol{V}\boldsymbol{c}, \ \tau \in (0, 1).$$

Since by definition $\boldsymbol{g}(\boldsymbol{y}) = \frac{\partial L(\boldsymbol{y})}{\partial \boldsymbol{y}}$, the expression above can be presented as follows

$$L(\boldsymbol{y} + \boldsymbol{V}\boldsymbol{c}) = L(\boldsymbol{y}) + (\boldsymbol{g}(\boldsymbol{y} + \tau\boldsymbol{V}\boldsymbol{c}))^\top \boldsymbol{V}\boldsymbol{c}, \ \tau \in (0, 1).$$

Now, since $\tau > 0$ we use $\boldsymbol{g}(\boldsymbol{y} + \tau\boldsymbol{V}\boldsymbol{c}) = \boldsymbol{g}(\boldsymbol{y})$ to remove shift and obtain

$$L(\boldsymbol{y} + \boldsymbol{V}\boldsymbol{c}) = L(\boldsymbol{y}) + (\boldsymbol{g}(\boldsymbol{y}))^\top \boldsymbol{V}\boldsymbol{c}$$

We substitute this to the last expression for energy which gives

$$E(\boldsymbol{y} + \boldsymbol{V}\boldsymbol{c}) = (\boldsymbol{y} + \boldsymbol{V}\boldsymbol{c} - \boldsymbol{b})^\top \boldsymbol{g}(\boldsymbol{y}) - L(\boldsymbol{y}) - (\boldsymbol{g}(\boldsymbol{y}))^\top \boldsymbol{V}\boldsymbol{c} - \frac{1}{2}(\boldsymbol{g}(\boldsymbol{y}))^\top \boldsymbol{W}\boldsymbol{g}(\boldsymbol{y}).$$

Two remaining terms containing $\boldsymbol{c}$ are $(\boldsymbol{V}\boldsymbol{c})^\top \boldsymbol{g}(\boldsymbol{y})$ and $-(\boldsymbol{g}(\boldsymbol{y}))^\top \boldsymbol{V}\boldsymbol{c}$ cancel each other, so we obtain

$$E(\boldsymbol{y} + \boldsymbol{V}\boldsymbol{c}) = (\boldsymbol{y} - \boldsymbol{b})^\top \boldsymbol{g}(\boldsymbol{y}) - L(\boldsymbol{y}) - \frac{1}{2}(\boldsymbol{g}(\boldsymbol{y}))^\top \boldsymbol{W}\boldsymbol{g}(\boldsymbol{y}) = E(\boldsymbol{y}),$$

as claimed in the proposition.

## C  EXAMPLE OF MEMORY CAPACITY DEGRADATION FOR ENERGY FUNCTION BOUNDED FROM BELOW

Figure 2a shows an energy function that is unbounded from below but supports two stable states. In this section we argue that the unbounded energy is not problematic. Moreover, if one is too strict about this property, memory can severely degrade in capacity.

It is easy to find papers on associative memory that claim that it is necessary to have energy function bounded from below to ensure stable memory recovery, e.g., see discussion after equation (4) (in both cases) in Krotov (2021) and Krotov & Hopfield (2020). In Xie & Seung (2003) authors even incorrectly claim "Furthermore, with appropriately chosen $f_k$, such as sigmoid functions, $E(x)$ is also bounded below, in which case $E(x)$ is a Lyapunov function". Clearly, the Lyapunov function is not required to be bounded.

We certainly agree that if the energy function is bounded and informative (not flat), the memory vector will be recovered eventually from arbitrary initial conditions. However, we have several arguments again the overall importance of this condition:

1. **Unbounded at infinity only.** Activation functions used for associative memory are at least continuous, meaning energy can be unbounded only when the activities of neurons reach infinity. This kind of run-away behavior is very easy to detect and prevent when one uses actual implementation of associative memory.

2. **Finite number of updates in practice.** In practice ODE is not integrated for a very long time. As we discuss in Section 5 one usually performs a small number of steps for discretised dynamics, e.g., 10 or even 1 step. Under these conditions it is not possible to diverge using ODEs considered in this article.

3. **Stability is a local condition.** For associative memory one cares more about local stability in the basins of attraction, and the fact that energy is bounded from below is a global information, which is seldom important for local stability (with the important exception discussed below). In practice one can always rescale input vectors (initial conditions) to localize it on a sufficiently small sphere with radius $R$. Memory trained and used this way has little chance to diverge.

4. **Extra constraints can compromise capacity.** If we require energy to be bounded from below we will put extra constraints on parameters of associative memory. These constraints, as we will show below, can decrease capacity dramatically.

The last point, the decrease in capacity, is especially interesting. We will illustrate it with ReLU memory model

$$\frac{d\boldsymbol{y}(t)}{dt} = \boldsymbol{W}\text{ReLU}\left(\boldsymbol{y}(t)\right) - \boldsymbol{y}(t) + \boldsymbol{b}, \, \boldsymbol{y}(0) = \boldsymbol{y}_0,$$

that has energy

$$E(\boldsymbol{y}) = (\boldsymbol{y} - \boldsymbol{b})^\top \text{ReLU}(\boldsymbol{y}) - \sum_{i=1}^{N} \frac{1}{2}\left(\text{ReLU}(y_i)\right)^2 - \frac{1}{2}\left(\text{ReLU}(\boldsymbol{y})\right)^\top \boldsymbol{W}\,\text{ReLU}(\boldsymbol{y}).$$

To analyze this energy function we observe that $\boldsymbol{y}^\top \text{ReLU}(\boldsymbol{y}) = \left(\text{ReLU}(\boldsymbol{y})\right)^\top \text{ReLU}(\boldsymbol{y})$. This fact allows us to simplify energy to

$$E(\boldsymbol{y}) = \frac{1}{2}\left(\text{ReLU}(\boldsymbol{y})\right)^\top (\boldsymbol{I} - \boldsymbol{W})\,\text{ReLU}(\boldsymbol{y}) - \boldsymbol{b}^\top \text{ReLU}(\boldsymbol{y}).$$

It is possible to bound this energy function from below using spectral radius $\rho$ of $\boldsymbol{W}$

$$E(\boldsymbol{y}) \geq \frac{1 - \rho(\boldsymbol{W})}{2}\left(\text{ReLU}(\boldsymbol{y})\right)^\top \text{ReLU}(\boldsymbol{y}) - \boldsymbol{b}^\top \text{ReLU}(\boldsymbol{y}),$$

note that this bound is tight unless we restrict allowed weights, since it is saturated for $\boldsymbol{W} = \alpha\boldsymbol{I}$ for scalar $\alpha$. From this lower bound we can observe that memory is bounded from below if $\rho(\boldsymbol{W}) \leq 1$. Moreover, in light of our previous comment, it is a necessary and sufficient condition.

Suppose now we restrict $\boldsymbol{W}$ such that $\rho(\boldsymbol{W}) = 1 - \epsilon$ with arbitrary small but nonzero $\epsilon$. In this case ReLU memory has a single memory vector.

We start by showing that a steady state of temporal dynamics exists. For that we consider a discrete iteration

$$\boldsymbol{y}^{(n+1)} = \boldsymbol{W}\text{ReLU}\left(\boldsymbol{y}^{(n)}\right) + \boldsymbol{b}.$$

Observe that right-hand side is a Lipschitz function with Lipschitz constant $1 - \epsilon$:

$$\left\|\boldsymbol{W}\text{ReLU}\left(\boldsymbol{c_1}\right) + \boldsymbol{b} - \left(\boldsymbol{W}\text{ReLU}\left(\boldsymbol{c_2}\right) + \boldsymbol{b}\right)\right\|_2 = \left\|\boldsymbol{W}\left(\text{ReLU}\left(\boldsymbol{c_1}\right) - \text{ReLU}\left(\boldsymbol{c_2}\right)\right)\right\|_2$$
$$\leq \left\|\boldsymbol{W}\right\|_2 \left\|\left(\text{ReLU}\left(\boldsymbol{c_1}\right) - \text{ReLU}\left(\boldsymbol{c_2}\right)\right\|_2 \leq \left\|\boldsymbol{W}\right\|_2 \left\|\boldsymbol{c_1} - \boldsymbol{c_2}\right\|_2 = (1 - \epsilon)\left\|\boldsymbol{c_1} - \boldsymbol{c_2}\right\|_2$$

Given that, by Banach fixed point theorem Palais (2007) these iterations has unique fixed point. This fixed point is a steady state since

$$\boldsymbol{y}^\star = \boldsymbol{W}\text{ReLU}\left(\boldsymbol{y}^\star\right) + \boldsymbol{b} \Rightarrow \boldsymbol{W}\text{ReLU}\left(\boldsymbol{y}^\star\right) - \boldsymbol{y}^\star + \boldsymbol{b} = 0$$

so the right-hand side of the dynamical system is zero.

Now, we will show that this steady state is unique. For that we consider two trajectories starting from (arbitrary) distinct initial conditions $\boldsymbol{y}_1(t), \boldsymbol{y}_2(t)$: $\boldsymbol{y}_1(0) = \boldsymbol{y}_0^1, \boldsymbol{y}_2(0) = \boldsymbol{y}_0^2$. We will show that

the distance between $\boldsymbol{y}_1(t)$ and $\boldsymbol{y}_2(t)$ decreases with time. To do that we consider derivative of this distance:

$$\frac{1}{2}\frac{d}{dt}\|\boldsymbol{y}_1(t) - \boldsymbol{y}_2(t)\|_2^2 = (\boldsymbol{y}_1(t) - \boldsymbol{y}_2(t))^\top \left(\frac{d\boldsymbol{y}_1(t)}{dt} - \frac{d\boldsymbol{y}_2(t)}{dt}\right)$$

$$= (\boldsymbol{y}_1(t) - \boldsymbol{y}_2(t))^\top (\boldsymbol{W}\text{ReLU}\,(\boldsymbol{y_1}(t)) - \boldsymbol{y_1}(t) - \boldsymbol{W}\text{ReLU}\,(\boldsymbol{y_2}(t)) + \boldsymbol{y_2}(t))$$

$$= -\|\boldsymbol{y}_1(t) - \boldsymbol{y}_2(t)\|_2^2 + (\boldsymbol{y}_1(t) - \boldsymbol{y}_2(t))^\top (\boldsymbol{W}\text{ReLU}\,(\boldsymbol{y_1}(t)) - \boldsymbol{W}\text{ReLU}\,(\boldsymbol{y_2}(t)))\,.$$

To bound second term we use that $a \le |a|$ and Cauchy-Schwarz inequality

$$\frac{1}{2}\frac{d}{dt}\|\boldsymbol{y}_1(t) - \boldsymbol{y}_2(t)\|_2^2 \le -\|\boldsymbol{y}_1(t) - \boldsymbol{y}_2(t)\|_2^2 + \left|(\boldsymbol{y}_1(t) - \boldsymbol{y}_2(t))^\top (\boldsymbol{W}\text{ReLU}\,(\boldsymbol{y_1}(t)) - \boldsymbol{W}\text{ReLU}\,(\boldsymbol{y_2}(t)))\right|$$

$$\le -\|\boldsymbol{y}_1(t) - \boldsymbol{y}_2(t)\|_2^2 + \|\boldsymbol{y}_1(t) - \boldsymbol{y}_2(t)\|_2 \|\boldsymbol{W}\text{ReLU}\,(\boldsymbol{y_1}(t)) - \boldsymbol{W}\text{ReLU}\,(\boldsymbol{y_2}(t))\|_2\,.$$

Finally, we use result that $\boldsymbol{W}\text{ReLU}(\cdot)$ is Lipschitz with Lipschitz constant $1 - \epsilon$ to obtain

$$\frac{1}{2}\frac{d}{dt}\|\boldsymbol{y}_1(t) - \boldsymbol{y}_2(t)\|_2^2 \le -\|\boldsymbol{y}_1(t) - \boldsymbol{y}_2(t)\|_2^2 + \|\boldsymbol{y}_1(t) - \boldsymbol{y}_2(t)\|_2^2\,(1 - \epsilon) = -\epsilon\|\boldsymbol{y}_1(t) - \boldsymbol{y}_2(t)\|_2^2\,.$$

Using Grönwal inequality we obtain

$$\frac{1}{2}\frac{d}{dt}\|\boldsymbol{y}_1(t) - \boldsymbol{y}_2(t)\|_2^2 \le e^{-2\epsilon t}\frac{1}{2}\frac{d}{dt}\|\boldsymbol{y}_1(0) - \boldsymbol{y}_2(0)\|_2^2\,.$$

In other words, separated trajectories converge exponentially fast. Since there is a fixed point one of the trajectories might as well start from $\boldsymbol{y}^\star$ which means this fixed point is the only one, and it is exponentially stable.

The result that we demonstrate here means one should be careful restricting parameters of associative memory, since it may lead to degradation of capacity to a single memory. The proof above is not working for $\rho(\boldsymbol{W}) = 1$, but since $\epsilon$ can be arbitrary slow one can recover the same result in a limit $\epsilon \to 0$.

## D    MEMORY MODEL FROM HOPFIELD (1984)

In this section we align our notation with the one from Hopfield (1984). The model of interest is given in (Hopfield, 1984, Equation (5)). We replicated this model below

$$C_i\frac{du_i}{dt} = \sum_j T_{ij}V_j - u_i/R_i + I_i,$$

$$u_i = g_i^{-1}\,(V_i)\,,$$

where $u_i$ is an instantaneous input to neuron $i$, $V_i$ is the output or "short term average of the firing rate of the cell $i$", $g_i$ is an activation function or "the input-output characteristic of a nonlinear amplifier with negligible response time", $T_{ij}$ are weights and $T_{ij}^{-1}$ "finite impedance between the output $V_j$ and the cell body of cell $i$", $R_i$ is "transmembrane resistance" and $C_i$ is "input capacitance", $I_i$ is bias or "any other (fixed) input current to neuron $i$".

If we put aside the biological content of Hopfield (1984), this equation is already very similar to all models we discussed in Appendix A. To make the resemblance even more evident observe that $R_i$ can be removed without the loss of generality, since we can multiply on them and redefine $C_i \to R_i C_i$, $T_{ij} \to R_i T_{ij}$, $I_i \to R_I I_i$. After that we also substitute $V_i = g(u_i)$ and obtain the following equation

$$C_i\frac{du_i}{dt} = \sum_j T_{ij}g_j(u_j) - u_i + I_i,$$

which is precisely the Dense Associative Memory (Krotov, 2021, Equation (2)) with weights $T_{ij}$, time variables $C_i$ extra bias term $I_i$. Since we already discussed the relation of Dense Associative Memory to our model in Appendix A, we can be sure that model from Hopfield (1984) is in the same class of memory models.

Dynamical system (Hopfield, 1984, Equation (5)) comes with the energy function (Hopfield, 1984, Equation (7)):

$$E = -\frac{1}{2}\sum_{ij}T_{ij}V_iV_j + \sum_i \frac{1}{R_i}\int_0^{V_i}dV\,g_i^{-1}(V) + \sum_i I_iV_i,$$

Using $V_i = g_i(u_i)$ we obtain

$$E = -\frac{1}{2}\sum_{ij}T_{ij}g_i(u_i)g_i(u_i) + \sum_i \frac{1}{R_i}\int_0^{V_i}dV\,g_i^{-1}(V) + \sum_i I_ig_i(u_i).$$

In addition to that we can drop $R_i$ for the reason explained above

$$E = -\frac{1}{2}\sum_{ij}T_{ij}g_i(u_i)g_i(u_i) + \sum_i \int_0^{V_i}dV\,g_i^{-1}(V) + \sum_i I_ig_i(u_i).$$

The first and last sums are easily mapped on our notation and the second sum is explained in Section 3.1).

All in all, one can say that our model is similar to the one from Hopfield (1984) but with $C_i = R_i = 1$. As we argued, $R_i$ is irrelevant and $C_i$ does not appear in the energy function.

## E  PROOF OF PROPOSITION 3: STABILISATION OF DYNAMICS FOR RANDOM MATRICES

Matrix $\boldsymbol{W}_\perp$ can be written as $\boldsymbol{D}^\top\boldsymbol{O}^\top\boldsymbol{W}\boldsymbol{O}\boldsymbol{D}$ where $\boldsymbol{O}$ is orthogonal full rank matrix and $\boldsymbol{D}$ is first $N - k$ columns of $\boldsymbol{I}$. Since GOE is invariant under orthogonal transformations $\boldsymbol{O}^\top\boldsymbol{W}\boldsymbol{O}$ is also a matrix from GOE. The effect of $\boldsymbol{D}$ is to select a principal submatrix with $N - k$ rows and columns. Given that $\boldsymbol{W}_\perp$ is also from GOE but with $N - k$ rows and columns. For GOE with large $N$, the distribution of leading eigenvalue quickly converges to Tracy–Widom distribution (see Chiani (2014) equation (50) and numerical results), so, on the average, spectral radius of a matrix with $N - k$ rows and columns is $\sqrt{2(N-k)}$ (see below) and $\mathbb{E}\,\|\boldsymbol{W}_\perp\|_2\,/\mathbb{E}\,\|\boldsymbol{W}\|_2 = \sqrt{1 - \frac{k}{N}}$.

To obtain an average spectral radius for the GOE matrix $\boldsymbol{M}$ we will use the theory described in Section 4 of Chiani (2014). Let $\lambda_1$ be a spectral radius of matrix $\boldsymbol{M} \in \mathbb{R}^{N\times N}$. We define a Gaussian orthogonal ensemble as a set of random symmetric matrices with independently identically normally distributed entries (for upper diagonal part) with variances $1$ and $1/2$ for entries on and off the diagonal respectively. For sufficiently large $N$ it is known that

$$\overline{\lambda}_1 = \mu_N' + \mu_{\mathrm{TW}_1}\sigma_N', \tag{11}$$

where $\overline{\lambda}_1$ is a mean value of spectral radius, $\mu_{\mathrm{TW}_1} \simeq -1.2$ is a mean value of Tracy–Widom distribution and

$$\mu_N' = \sqrt{2\left(N - \frac{1}{2} - \frac{1}{10}\left(N - \frac{1}{2}\right)^{-\frac{1}{3}}\right)} \to \sqrt{2N-1};\ \sigma_N' = \frac{1}{\sqrt{2}}N^{-\frac{1}{6}} \to 0. \tag{12}$$

So we see that for large $N$ the average spectral radius of the GOE matrix is $\sqrt{2N}$.

## F  PROOF OF PROPOSITION 5: ASSOCIATIVE MEMORY WITHOUT DEAD NEURONS

Observe that $\frac{\partial L(\boldsymbol{W}\boldsymbol{u}+\boldsymbol{b})}{\partial\boldsymbol{u}} = \boldsymbol{W}\boldsymbol{g}\left(\boldsymbol{W}\boldsymbol{u}+\boldsymbol{b}\right)$ and $\frac{\partial\boldsymbol{g}(\boldsymbol{W}\boldsymbol{u}+\boldsymbol{b})}{\partial\boldsymbol{u}} = \boldsymbol{\Lambda}\left(\boldsymbol{W}\boldsymbol{u}+\boldsymbol{b}\right)\boldsymbol{W}$.

1. Using the identities above we find

$$\frac{\partial E_1}{\partial\boldsymbol{u}} = \boldsymbol{W}\boldsymbol{\Lambda}\left(\boldsymbol{W}\boldsymbol{u}+\boldsymbol{b}\right)\boldsymbol{W}\left(\boldsymbol{u}-\boldsymbol{g}(\boldsymbol{W}\boldsymbol{u}+\boldsymbol{b})\right),\ \frac{\partial E_2}{\partial\boldsymbol{u}} = \boldsymbol{W}\left(\boldsymbol{u}-\boldsymbol{g}(\boldsymbol{W}\boldsymbol{u}+\boldsymbol{b})\right),$$

$$\frac{\partial E_3}{\partial\boldsymbol{u}} = \left(\boldsymbol{I}-\boldsymbol{W}\boldsymbol{\Lambda}\left(\boldsymbol{W}\boldsymbol{u}+\boldsymbol{b}\right)\right)\boldsymbol{S}\left(\boldsymbol{u}-\boldsymbol{g}(\boldsymbol{W}\boldsymbol{u}+\boldsymbol{b})\right).$$

Multiplying gradients on $\alpha, \beta, \gamma$ and adding them gives $\frac{\partial E(\boldsymbol{u})}{\partial \boldsymbol{u}} = \boldsymbol{F}(\boldsymbol{u})\boldsymbol{f}(\boldsymbol{u})$. Since $\dot{\boldsymbol{u}} = -\boldsymbol{R}(\boldsymbol{u})\boldsymbol{f}(\boldsymbol{u}(t))$ where $\boldsymbol{f}(\boldsymbol{u}(t)) = \boldsymbol{u} - \boldsymbol{g}(\boldsymbol{W}\boldsymbol{u}+\boldsymbol{b})$ from $\dot{E} = \dot{\boldsymbol{u}}^\top \frac{\partial E}{\partial \boldsymbol{u}}$ we obtain

$$\dot{E} = -\boldsymbol{f}(\boldsymbol{u})^\top \left( \boldsymbol{R}^\top \boldsymbol{F}(\boldsymbol{u}) + (\boldsymbol{F}(\boldsymbol{u}))^\top \boldsymbol{R} \right) \boldsymbol{f}(\boldsymbol{u}).$$

If one can find positive semidefinite matrix $\boldsymbol{Q}$ such that $\boldsymbol{R}^\top \boldsymbol{F}(\boldsymbol{u}) + (\boldsymbol{F}(\boldsymbol{u}))^\top \boldsymbol{R} \geq \boldsymbol{Q}$, energy is non-increasing on trajectories since in this case $\dot{E} \leq -\boldsymbol{f}^\top \boldsymbol{Q} \boldsymbol{f} \leq 0$

2. Consider polar decomposition of matrix $\boldsymbol{F}(\boldsymbol{u}) = \boldsymbol{O}(\boldsymbol{u})\boldsymbol{P}(\boldsymbol{u})$ and take $\boldsymbol{R}(\boldsymbol{u}) = \boldsymbol{O}(\boldsymbol{u})$. Since $\boldsymbol{P}(\boldsymbol{u}) \geq 0$ we have $\boldsymbol{R}^\top \boldsymbol{F}(\boldsymbol{u}) + (\boldsymbol{F}(\boldsymbol{u}))^\top \boldsymbol{R} = 2\boldsymbol{P}(\boldsymbol{u}) \geq 0$ and $\dot{E} \leq 0$.

3. From the proof of Proposition 1 we know that if $\boldsymbol{W}\boldsymbol{u} + \boldsymbol{b} \to \boldsymbol{W}\boldsymbol{u} + \boldsymbol{b} + \boldsymbol{V}\boldsymbol{c}$, Lagrange function shifts on $\boldsymbol{g}(\boldsymbol{W}\boldsymbol{u} + \boldsymbol{b})\boldsymbol{V}\boldsymbol{c}$. The first term in the energy function (6) is quadratic and does not contain $\boldsymbol{g}$ so in general energy $E_2$ does not have flat directions. Similarly, $E_3$ has a form $\boldsymbol{f}(\boldsymbol{u})^\top \boldsymbol{S} \boldsymbol{f}(\boldsymbol{u})$ where $\boldsymbol{f}$ is not invariant under transformation $\boldsymbol{W}\boldsymbol{u} + \boldsymbol{b} \to \boldsymbol{W}\boldsymbol{u} + \boldsymbol{b} + \boldsymbol{V}\boldsymbol{c}$ meaning $E_3$ is not invariant as well.

4. Since $\frac{\partial E}{\partial \boldsymbol{u}} = \boldsymbol{F}(\boldsymbol{u})\boldsymbol{f}(\boldsymbol{u})$ and $\boldsymbol{f}(\boldsymbol{u}^\star) = 0$ we find for sufficiently small $\boldsymbol{\delta}$

$$E(\boldsymbol{u}^\star + \boldsymbol{\delta}) - E(\boldsymbol{u}^\star) = \boldsymbol{\delta}^\top \boldsymbol{F}(\boldsymbol{u}^\star) \left.\frac{\partial \boldsymbol{f}(\boldsymbol{u})}{\partial \boldsymbol{u}}\right|_{\boldsymbol{u}=\boldsymbol{u}^\star} \boldsymbol{\delta}.$$

For stability one needs this quadratic form to be positive definite, which gives (8).

