# OpenReview forum: "Associative memory and dead neurons"
_ICLR.cc/2025/Conference — ICLR 2025 Poster_

### Official Review · Reviewer_WSRe · 2024-10-23

**Soundness:** 2
**Presentation:** 3
**Contribution:** 3
**Rating:** 6
**Confidence:** 3

**Summary:**

This paper builds on the work of Krotov and Hopfield 2020 looking at associative memories, with particular emphasis on the classical continuous Hopfield network (Hopfield 1984), the Dense Associative Memory (Krotov and Hopfield 2016), the modern continuous Hopfield network (Ramsauer et al. 2020), and the energy transformer (Hoover et al. 2023). More broadly, this paper looks at associative memories as dynamical systems, similar to neural ODEs, and investigates the impact of dead neurons on such systems. A dead neuron in this context refers to a neuron which has no effect on the model behavior due to always taking values that are in some way constant. Dead neurons can arise from many situations, but two such classes discussed in Section 2 are saturated neurons and symmetric activation functions. This paper then analyzes how dead neurons impact the degrees of freedom of a dynamical system and the Lyapunov function of the models. In particular, it is shown that the associative memories in current literature may suffer from steady states that are not stable i.e. there are dimensions of the state space (associated with each dead neuron) that are free to vary without affecting the energy. The paper then introduces new energy functions that alleviates some of the issues relating to dead neurons and shows all architectures explained by the functions in Krotov and Hopfield 2020 are also explainable using the new functions by an affine transformation of the neuron activities.

**Strengths:**

The work seems rigorous in its discussions, providing proofs of propositions and analyzing the proposed energy functions extensively. The work appears original, although it is mostly building off previous associative memory literature and work on dead neurons. The paper appears to be reasonably significant in the field of associative memories, and if the claims of the impact of dead neurons are true, this could represent a significant step in the analysis of neural ODEs and similar dynamical systems.

**Weaknesses:**

The basis of the paper, Section 2, is confusing and difficult to match up to other literature on associative memories. In particular, Equations 1 through 4, which are fundamental to the remainder of the paper, are stated to be from Krotov and Hopfield 2020, but do not appear in the form given in this paper. A rewriting of Section 2 would be beneficial to improve the clarity of the work and aid reader's understanding. Rewriting of subsequent sections to reference a clarified Section 2 would also help a reader follow each point made by this work.

The notation used in the paper is unique to other papers on associative memories, which in itself is not an issue, but there is no explanation of the notation. This is a problem when discussing other literature, as matching the discussions in this paper to other referenced papers becomes nearly impossible. In many sections, steps are taken that are not obvious to the reader and may impact the ability to follow this work. There are numerous grammatical errors throughout the manuscript that can obfuscate the meaning of sentences or entire paragraphs. It is difficult to verify the claims made in this paper due to the difficult-to-follow nature of each section.

The key ideas of the paper seem sufficient for publication, but would need to be presented in a more understandable format.

**Questions:**

There seems to be a typo on line 044: "The most cluical requirement" should be "The most crucial requirement".

Equations 1 through 4 --- what do these correspond to in Krotov and Hopfield 2020? Looking through the cited paper, there are no equations of a similar form, have the authors of this paper rewritten something from Krotov and Hopfield 2020 in a more convenient form? If so, perhaps a brief description of what has been changed from Krotov and Hopfield 2020 (a notation translation guide) would benefit this paper. Further, Equations 1 through 4 appear to actually consist of more than four equations, some of which are constraints (such as W = W transpose). Cleaning up the formatting here such that each equation is on one line with a separate number may benefit the paper.

At the start of Section 2 the paper makes several claims about the properties of Equations 1 through 4, including important properties of the energy function in Equation 4. However, these claims are not obvious to me. Is there a reason that the "structural assumptions" of Equation 2 (is that the first or second equation on line two?) ensures the energy function exists? This discussion could be expanded slightly for readers who are not familiar with the equations set forth here.

The next paragraph continues these discussions and makes claims about the dynamical systems described by Equation 1. However, again I can not immediately see why the steady states of these differential equations can be used as memory vectors --- could this be discussed further or given a citation?

Example 1 seems somewhat ill-defined in the edge cases. The given differential equation for y_i does not apply for y_1; what is the value of the weight matrix W_{10}? Should we treat this as 0? This would line up with my expectation of the MLP with feedback connections, but the formalization does not support it. Perhaps some discussion of the edge cases here would be beneficial.

There is a slight typesetting error on line 156: The equation is currently rendered like "$ W_{12} = W_{2}1^\top $", rather than "$ W_{12} = W_{21}^\top $" i.e. the "1" on the second matrix W is not subscript.

Example 2 is the first time that the term "dead neuron" is linked directly to the ReLU activation. Dead neurons are referenced to Lu et al. 2019 in the introduction (line 091) but perhaps a short introduction to the term to ground this work would help a reader understand where the discussion is heading. Especially considering the term is then applied to other activations in Examples 3 and 4. What exactly is meant by a neuron being dead? This could be added to the introduction. Reading on, the definition of a dead neuron appears quite late in Section 2, which is already substantially through the paper. Is there some way to introduce the key concepts of the paper earlier?

The paper seems to have a significant focus on the energy function being non-discriminative on a non-compact region of state space. It may benefit a reader's understanding to have a short discussion on what this means, especially in the context of associative memories and physics inspired models.

The proof for Proposition 1 is somewhat difficult to follow. The Taylor expansion of the Lagrangian seems to make the implicit assumption that $ L(y) = 0.5 g(y)^2 $, which perhaps could be stated explicitly in the proof even though this is a common formulation. Past this, it is not clear to me why the second term shifts by the specified quantity, and how this is compensated by a similar shift in the first term. Some more verbose discussion here could help readers follow the proof.

Much of the body of this paper hinges on the formalization of Equations 1-4. Without having this section very clearly laid out for the reader, the remainder of the paper is often difficult to follow. Is it possible to rework this initial Section 2 to improve clarity? In particular, stating notation clearly early in the paper would be very helpful. Following sections would then be able to reference back to the key statements in Section 2 with minimal confusion.

Again for Equation 5 it is not clear what Lyapunov function you are referencing in Hopfield 1984. This looks to be most similar to Equation 7 or 11 in the 1984 paper, but the lack of notation formalization in this work makes it difficult to draw parallels.

Much of Section 3 and 4, (e.g. Section 3.2), may benefit from being written into a longer paper where the authors can be more verbose in explaining each step of the mathematics.  Currently, the paper is very terse in its discussion, which is understandable considering the page limit of the conference, but this does not help a reader understand the key aspects of this mathematically dense manuscript.

Near the start of Section 4, where we discuss the value of R(u), are we assuming that no value of the matrix R(u) is zero, or that the matrix is not equal to the zero matrix (i.e. all values zero)? The manuscript is slightly unclear on this.

---

> ### Author Response · Authors · 2024-11-20
>
> ```
> There seems to be a typo on line 044: "The most cluical requirement" should be "The most crucial requirement".
> ```
> We corrected this typo.

---

> ### Author Response · Authors · 2024-11-20
>
> ```
> The basis of the paper, Section 2, is confusing and difficult to match up to other literature on associative memories. In particular, Equations 1 through 4, which are fundamental to the remainder of the paper, are stated to be from Krotov and Hopfield 2020, but do not appear in the form given in this paper. A rewriting of Section 2 would be beneficial to improve the clarity of the work and aid reader's understanding. Rewriting of subsequent sections to reference a clarified Section 2 would also help a reader follow each point made by this work.
> ```
> also
> ```
> Equations 1 through 4 --- what do these correspond to in Krotov and Hopfield 2020? Looking through the cited paper, there are no equations of a similar form, have the authors of this paper rewritten something from Krotov and Hopfield 2020 in a more convenient form? If so, perhaps a brief description of what has been changed from Krotov and Hopfield 2020 (a notation translation guide) would benefit this paper. Further, Equations 1 through 4 appear to actually consist of more than four equations, some of which are constraints (such as W = W transpose). Cleaning up the formatting here such that each equation is on one line with a separate number may benefit the paper.
> ```
> also
> ```
> Much of the body of this paper hinges on the formalization of Equations 1-4. Without having this section very clearly laid out for the reader, the remainder of the paper is often difficult to follow. Is it possible to rework this initial Section 2 to improve clarity? In particular, stating notation clearly early in the paper would be very helpful. Following sections would then be able to reference back to the key statements in Section 2 with minimal confusion.
> ```
> We agree that the previous version of the paper may lead to confusion. The reason, as the reviewer correctly points out, is that memory models are plentiful and notation is not always aligned. To improve the situation we reworked the beginning of Section 2. In a new version we explicitly point to the appropriate literature and explain why each assumption is necessary.
>
> Besides, we arrange a new Appendix A which explicitly considers two main models and their relation to other models. There we show why "Dmitry Krotov and John Hopfield. Large associative memory problem in neurobiology and machine learning" is a particular case of "Dmitry Krotov. Hierarchical associative memory" and what is the relation to the model from "Dmitry Krotov. Hierarchical associative memory" and our model.

---

> ### Author Response · Authors · 2024-11-20
>
> ```
> At the start of Section 2 the paper makes several claims about the properties of Equations 1 through 4, including important properties of the energy function in Equation 4. However, these claims are not obvious to me. Is there a reason that the "structural assumptions" of Equation 2 (is that the first or second equation on line two?) ensures the energy function exists? This discussion could be expanded slightly for readers who are not familiar with the equations set forth here.
>
> The next paragraph continues these discussions and makes claims about the dynamical systems described by Equation 1. However, again I can not immediately see why the steady states of these differential equations can be used as memory vectors --- could this be discussed further or given a citation?
> ```
> In a revision the beginning of the Section 2 was reformulated to accommodate suggestions proposed by the reviewer. More specifically, we clarify why various assumptions are needed and provide references. We point to the introduction where we discuss why this kind of ODEs are suitable for storing memories and also supply references to the works of Krotov and Hopfield where these questions are extensively discussed. We also reference Appendix A where we discuss how our memory model is related to the old ones.

---

> ### Author Response · Authors · 2024-11-20
>
> ```
> Example 1 seems somewhat ill-defined in the edge cases. The given differential equation for y_i does not apply for y_1; what is the value of the weight matrix W_{10}? Should we treat this as 0? This would line up with my expectation of the MLP with feedback connections, but the formalization does not support it. Perhaps some discussion of the edge cases here would be beneficial.
> ```
> We agree that the edge cases were not considered carefully. In the revision we explicitly write them down.

---

> ### Author Response · Authors · 2024-11-20
>
> ```
> There is a slight typesetting error on line 156: The equation is currently rendered like "$ W_{12} = W_{2}1^\top $", rather than "$ W_{12} = W_{21}^\top $" i.e. the "1" on the second matrix W is not subscript.
> ```
> This typo is corrected.

---

> ### Author Response · Authors · 2024-11-20
>
> ```
> Example 2 is the first time that the term "dead neuron" is linked directly to the ReLU activation. Dead neurons are referenced to Lu et al. 2019 in the introduction (line 091) but perhaps a short introduction to the term to ground this work would help a reader understand where the discussion is heading. Especially considering the term is then applied to other activations in Examples 3 and 4. What exactly is meant by a neuron being dead? This could be added to the introduction. Reading on, the definition of a dead neuron appears quite late in Section 2, which is already substantially through the paper. Is there some way to introduce the key concepts of the paper earlier?
> ```
> Indeed, the definition of dead neurons appeared too late in the previous version of the paper. We provide an additional footnote in the introduction that informally explains what we mean by dead neurons. This explanation includes both old uses (ReLU) and new ones (softmax, layer norm).

---

> ### Author Response · Authors · 2024-11-20
>
> ```
> The paper seems to have a significant focus on the energy function being non-discriminative on a non-compact region of state space. It may benefit a reader's understanding to have a short discussion on what this means, especially in the context of associative memories and physics inspired models.
> ```
> We extend discussion on that topic. Additional comments appear in a new Figure 2, in the end of Section 2, in appendix on page 7.

---

> ### Author Response · Authors · 2024-11-20
>
> ```
> The proof for Proposition 1 is somewhat difficult to follow. The Taylor expansion of the Lagrangian seems to make the implicit assumption that $ L(y) = 0.5 g(y)^2 $, which perhaps could be stated explicitly in the proof even though this is a common formulation. Past this, it is not clear to me why the second term shifts by the specified quantity, and how this is compensated by a similar shift in the first term. Some more verbose discussion here could help readers follow the proof.
> ```
> We did not make any implicit assumptions in this proof. The only assumptions are those needed to define the memory model, so the Lagrange function is not restricted to the form the reviewer suggested. We recognise that the proof was too short and likely confusing to many potential readers. In a revised version we expand this proof substantially and put it in Appendix A.

---

> ### Author Response · Authors · 2024-11-20
>
> ```
> Again for Equation 5 it is not clear what Lyapunov function you are referencing in Hopfield 1984. This looks to be most similar to Equation 7 or 11 in the 1984 paper, but the lack of notation formalization in this work makes it difficult to draw parallels.
> ```
> We agree that this part may, again, confuse the reader, because the notation in the 1984 paper is different from the one used in this paper. In the revised version we clarify the relation between different notations in Appendix D.

---

> ### Author Response · Authors · 2024-11-20
>
> ```
> Much of Section 3 and 4, (e.g. Section 3.2), may benefit from being written into a longer paper where the authors can be more verbose in explaining each step of the mathematics. Currently, the paper is very terse in its discussion, which is understandable considering the page limit of the conference, but this does not help a reader understand the key aspects of this mathematically dense manuscript.
> ```
> We understand that discussion can be dense in those sections. The page limit certainly does not help us here, but we are not going to blame ICLR constraints. We reworked text in parts and put several new motivations and clarification in these sections.
>
> We also want to point the reviewer attention that Section 4 now has the following structure:
> 1. Section opens with non-technical motivation why one need to consider modifications of dynamics
> 2. Next one can find a simplified example motivated from the theory of conservative fields.
> 3. After that the main parametric form of ODEs is introduced
> 4. It follows with a single formal statement
> 5. This statement is followed by an explanation of its meaning in plain english.
> 6. After that we give three examples of this newly introduced memory.
>
> We think the reviewer would agree that it is not completely fair to say this format is too dense. We strive to provide extra clarifications to the reader, but we recognise that we are not always successful in this attempt.

---

> ### Author Response · Authors · 2024-11-20
>
> ```
> Near the start of Section 4, where we discuss the value of R(u), are we assuming that no value of the matrix R(u) is zero, or that the matrix is not equal to the zero matrix (i.e. all values zero)? The manuscript is slightly unclear on this.
> ```
>
> This is a very good comment, we thank the reviewer for this observation. Indeed, it was a sloppy part. In the revised manuscript it is replaced with a proper discussion where we explain that the main requirement on R(u) is that it does not lose information, that is, it has empty nullspace.

---

> ### Author Response · Authors · 2024-11-20
>
> Dear WSRe,
> We find your review very useful, since it allows us to clarify many points that would remain obscure otherwise. We suggest you get familiar with a revised manuscript and come back with additional comments. If appropriate, you may also consider revising your score after that.

---

> > ### Comment · Reviewer_WSRe · 2024-11-21
> > **Response to Authors**
> >
> > I thank the authors for their immense revisions. Although much of the mathematics still exceeds my ability to rigorously scrutinize, the broad strokes are now much clearer to me and show very interesting results. The additional descriptions and discussions the authors have included have made the manuscript much more approachable for the average reader. Moving the vast majority of proofs to appendices has also given the paper a better flow, which again improves readability. From these substantial revisions I now believe this paper has suitable advancements for the field of associative memories *and* is approachable enough for researchers in said field. I have updated my review accordingly. From my reading of the updated manuscript I am satisfied with the current form, and further comments are likely superfluous.

---

### Official Review · Reviewer_oj8s · 2024-11-02

**Soundness:** 3
**Presentation:** 2
**Contribution:** 3
**Rating:** 6
**Confidence:** 2

**Summary:**

This work expands on the literature of modern Hopfield networks, and that of Dense associative memories. In detail, it identifies a drawbacks of the current approach developed by Krotov and Hopfield, that is, the presence of ‘dead neurons’ that cause regions of constant energy. The authors then address this problem by defining a new family of Lyapunov functions that is a variation of the original one, but does not suffer from the problem of dead neurons causing flat regions.

**Strengths:**

The work is well structured: it identifies a gap in the literature, and theoretically addresses it. The gap is also a significant one, as flat regions of steady states can negatively affect the retrieval of specific memories.

The stability analysis is interesting, and it seems well made. Similarly, I share the belief of the authors about the underutilisation of the Lyapunov function in the literature. Actually, I would move the discussion in the conclusion (“*The role of Lyapunov’s function is merely to provide comfort that some steady states may exist somewhere. As we show in this article, it is often a false comfort.*”) in the introduction, to make the claim of the paper clearer.

Despite not being sure I was able to follow the math all the way, I believe the solution proposed by the authors to be correct.

**Weaknesses:**

The biggest cons of this work is the clarity. I found it hard to follow the computations, theorems, claims, and their consequences. The 10 pages are very dense. I would:

1) Add figures that make some concepts clearer;

2) Move some non-interesting proofs to the appendix, and leave in the main body only the computations needed to understand concepts (you could even provide proof sketches).

As I did not manage to follow the paper perfectly, I will also provide a low confidence score.

**Questions:**

Are there specific tasks in which you could measure the advantage of the newly proposed energy function? For example, testing how many stored memories can you perfectly retrieve, or the capacity of the model? Similarly to the many provided in Millidge et al.?

---

> ### Author Response · Authors · 2024-11-20
>
> ```
> The biggest cons of this work is the clarity. I found it hard to follow the computations, theorems, claims, and their consequences. The 10 pages are very dense. I would:
>
> 1. Add figures that make some concepts clearer;
> 2. Move some non-interesting proofs to the appendix, and leave in the main body only the computations needed to understand concepts (you could even provide proof sketches).
> ```
> We agree with the reviewer that the paper can be hard to follow. To alleviate this issue we added many extra clarifications (in all sections, including several new footnotes) in the main text. We also move proofs to the Appendices as the reviewer suggested. This allowed us to introduce additional illustrations that motivate the study of energy.

---

> ### Author Response · Authors · 2024-11-20
>
> ```
> Are there specific tasks in which you could measure the advantage of the newly proposed energy function? For example, testing how many stored memories can you perfectly retrieve, or the capacity of the model? Similarly to the many provided in Millidge et al.?
> ```
> The answer to this question is yes, our formalism allows more general memory models. This is briefly explained in Example 7 where we build a model with a non-symmetric weight matrix $\boldsymbol{W}$ and not a positive-definite Hessian of Lagrange function. This is impossible to do with other associative memory models, because for them weights are required to be symmetric to have a well-defined energy function and Hessian should be positive definite for the energy function to be non-increasing on the trajectories of the dynamical system.
>
> To give a concrete example, old memory models can not be used with the GELU activation function, but our new model can.

---

> ### Author Response · Authors · 2024-11-20
>
> Dear oj8s,
> We kindly ask you to look through the revised version and revise the score or come with additional suggestions.

---

> ### Comment · Reviewer_oj8s · 2024-11-21
>
> Dear authors,
> Thank you for your answer. I have gone through the manuscript and I believe the presentation has been largely improved. I will raise my score as it was my main concern.

---

### Official Review · Reviewer_yNbD · 2024-11-10

**Soundness:** 3
**Presentation:** 3
**Contribution:** 3
**Rating:** 8
**Confidence:** 4

**Summary:**

The work investigates Lyapunov functions and stability in the Krotov and Hopfield models of associative memory. It is found that the conventional formulation suffers from a problem of dead neurons, which arises when some of the neurons’ activation functions saturate. In this situation the energy becomes independent of the state of those neurons. The authors propose a projection of the original class of models on a subspace of non-dead neurons, analyze the stability of those models, and propose a novel class of Lyapunov functions that describe those models.

**Strengths:**

Dense Associative Memory models have several possible formulations and admit studies of a broad class of AI models used in practice. At the same time, their Lyapunov (energy) functions remain underutilized and the shape of those energy landscapes remains under investigated. This paper tackles this exact question and analyses the stability of the dynamics and the sensitivity of the energy with respect to the state of the neurons from a novel angle. The problem of dead neurons can be a serious obstacle for training these models at scale in interesting AI applications. This work formalizes this problem, and offers a possible solution by identifying non-dead subspaces and proposing a novel class of energy functions. The paper is generally well-written, although there are small issues, which I explain below. The results presented in this work are novel and relevant to the ICLR community.

**Weaknesses:**

There are two issues with the way how authors introduce the problem of dead neurons (example 2, 3, 4). First, in the Krotov-Hopfield formalism, one needs to check two things:
1. Positive definiteness of the Hessian of the Lagrangian.
2. Boundness of the energy function from below.

Only when both conditions are satisfied the models under consideration have stable fixed point dynamics. The authors rightfully focus on the first requirement in their presentation, but somehow the second one is not discussed enough. For instance, please notice that the model studied in Example 2 does not have an energy bounded from below. This means that one can choose a matrix $\mathbf{W}$ such that the energy will keep decreasing to minus infinity with time, and the model will never reach a fixed point. The lower-boundness of the energy needs to be checked and discussed in the paper.


Second, Example 4 represents a situation when Lagrangian has a symmetry. Please notice (this is acknowledged in lines 194-195) that in this case there are no dead neurons, all the neurons are alive, it’s just the energy function is insensitive to the states of the variables $\mathbf{y}$ along the constant shifts of all pre-activations. I am wondering to what extent it makes sense to dub this specific example as a “dead-neuron” problem. The authors rightfully acknowledge this in lines 192-198, but I would invite the authors to consider restructuring the presentation of this example and give this specific phenomenon (Example 4) a different name, e.g. “symmetry”, “flat directions”, etc., not the dead-neuron problem.

Some typos:
1. Line 507: $E_3(\mathbf{u})$ -> $E_3(\mathbf{y})$
2. Line 509: $\Lambda(\mathbf{u})$ -> $\Lambda(\mathbf{y})$
3. Line 156-157 - subscript in $W_{21}$.
4. Line 535: Lapynov-> Lyapunov

**Questions:**

1. I am not sure I understood the derivation in line 284. Why does the Taylor series terminate after the second term?
2. It seems to me that the model presented in equation 9 can be obtained from the Krotov-Hopfield model by introducing a new variable $\mathbf{y}=\mathbf{Wu}+\mathbf{b}$. Are these two classes of models equivalent under this change of variables, or does the introduction of matrix R make equation 9 more general than Krotov and Hopfield?
3. In all the derivations $\mathbf{b}$ is assumed to be time-independent. This is fine (same is true in Krotov-Hopfield). I am wondering though, if anything could be said about situations when $\mathbf{b}$ depends on time? E.g., could energy function (10) still guarantee convergences for some properties of $\mathbf{b(t)}$?

---

> ### Author Response · Authors · 2024-11-20
>
> ```
> There are two issues with the way how authors introduce the problem of dead neurons (example 2, 3, 4). First, in the Krotov-Hopfield formalism, one needs to check two things:
>
> 1. Positive definiteness of the Hessian of the Lagrangian.
> 2. Boundness of the energy function from below.
>
> Only when both conditions are satisfied the models under consideration have stable fixed point dynamics. The authors rightfully focus on the first requirement in their presentation, but somehow the second one is not discussed enough. For instance, please notice that the model studied in Example 2 does not have an energy bounded from below. This means that one can choose a matrix $\boldsymbol{W}$ such that the energy will keep decreasing to minus infinity with time, and the model will never reach a fixed point. The lower-boundness of the energy needs to be checked and discussed in the paper.
> ```
> We would like to thank the reviewer for this suggestion. Research work done before the preparation of the manuscript contained results on the conditions needed to bound energy from below. We decided to put these results aside because they did not fit well in the topic of the paper: the dead neurons. However, since the reviewer wants to know what we think, we find it appropriate to include the discussion and some of the results in Appendix C. They are also described in Figure 2 and in Section 2 of the main text.
>
> In short, we disagree that these conditions are important from both theoretical and practical perspectives.
>
> Practically, all functions used are continuous, so energy is always bounded unless we are infinitely far from the origin. This is easy to control in implementation. Moreover, given common practice to perform a fixed number of iterations of, say, Euler scheme, whether energy bounded or unbounded is completely irrelevant.
>
> On the theoretical side, one can show that for Example 2, i.e., ReLU associative memory, suggested by the reviewer, one can obtain necessary and sufficient conditions for energy function to be bounded from below. After that we show that if energy is bounded, associative memory can support a single memory vector. which is globally exponentially stable. This illustrates that bounding energy from below can be a bad idea, since it compromises memory capacity.
>
> Of course Energy is trivially bounded for functions with saturations, e.g., sigmoid. But we know that in this case all points with $\left\|y\right\|_{2} > R$ lay in a flat region for sufficiently large $R$, so energy like that is not informative far from the origin.

---

> > ### Comment · Reviewer_yNbD · 2024-11-24
> > **comment**
> >
> > It is of course true that the energy may be unbounded in some regions of the neuron activity space, and bounded in other regions. In these cases if the network is initialized in the regions where the energy is unbounded the dynamics can become unstable, and the use of associative memories can be compromised. One can stop the dynamics after some number of steps, as you propose in lines 760-763, but the output of the network will strongly depend on this arbitrary cutoff (the chosen number of steps).

---

> > > ### Author Response · Authors · 2024-11-25
> > >
> > > We agree that the solution with a cutoff is not ideal. The alternative, having energy bounded from below, can also be problematic.
> > >
> > > First, the conditions on the parameters can be too restrictive. As we show in Appendix C, for the ReLU network associative memory can support only a single stationary state when energy is bounded from below. This result also applies to ReLU networks with any number of layers, due to their connection to Hierarchical Associative Memory.
> > >
> > > Second, the control of the energy function at infinity is “biologically implausible”. When considering energy as a real quantity, not just a mathematical abstraction, it's unrealistic to require it to be bounded on the entire space  $\mathbb{R}^{N}$, including regions never encountered in the initial conditions.
> > >
> > > The best suggestion we have now is to restrict the set of initial conditions to a small fraction of states, i.e., $\\left\\|y(t=0)\\right\\|_2\\leq R$, and train the memory such that all points within the set $\\left\\{y:\left\\|y\right\\|_2\leq R\\right\\}$ belong to the basin of attraction of some memory vector $y_i: \\left\\|y_i\\right\\|_2\\leq\\infty$.
> > >
> > > The other interesting possibility is to replace the attractor with transients (https://arxiv.org/abs/2404.10369). However, this would be a major modification of the whole framework.

---

> ### Author Response · Authors · 2024-11-20
>
> ```
> Second, Example 4 represents a situation when Lagrangian has a symmetry. Please notice (this is acknowledged in lines 194-195) that in this case there are no dead neurons, all the neurons are alive, it’s just the energy function is insensitive to the states of the variables  along the constant shifts of all pre-activations. I am wondering to what extent it makes sense to dub this specific example as a “dead-neuron” problem. The authors rightfully acknowledge this in lines 192-198, but I would invite the authors to consider restructuring the presentation of this example and give this specific phenomenon (Example 4) a different name, e.g. “symmetry”, “flat directions”, etc., not the dead-neuron problem.
> ```
> We would like to respectfully disagree. It is true that one usually not claim that softmax lead to dead neurons. However, our definition of dead neurons is that it is not possible to recover input to activation function from the output, i.e., that the information is lost (this definition is added in the footnote in the Introduction). This is true for softmax function. Besides, we argue that softmax can be considered as function that is saturated at each point (this is added in the footnote on page 4). To see that, consider new basis $\boldsymbol{u}\_1, \dots, \boldsymbol{u}\_n$ in $\mathbb{R}^{n}$ such that $\left(\boldsymbol{u}\_1\right)_{i} = \frac{1}{\sqrt{n}}, i=1,\dots,n$ and all other vectors are chosen anyhow to form orthonormal complete set. If we consider input and output of softmax in this bases, we will observe that the first component for the input $\boldsymbol{u}\_1^\top\boldsymbol{y}$ always changes to $\boldsymbol{u}\_1^\top\text{softmax}\left(\boldsymbol{y}\right) = \frac{1}{\sqrt{n}}$. In other words this first neuron (in a new basis) is saturated.

---

> ### Author Response · Authors · 2024-11-20
>
> ```
> Some typos:
>
> Line 507: $E_3(\boldsymbol{u})$ -> $E_3(\boldsymbol{y})$
> Line 509:  ->
> Line 156-157 - subscript in $\boldsymbol{W}_{21}$.
> Line 535: Lapynov-> Lyapunov
> ```
> We thank the reviewers for spotting these typos, they are corrected in the revised version of the manuscript.

---

> ### Author Response · Authors · 2024-11-20
>
> ```
> I am not sure I understood the derivation in line 284. Why does the Taylor series terminate after the second term?
> ```
> Here we use the Taylor series with the mean-value reminder. The last derivative is evaluated not in $\boldsymbol{y}^{\star}$. But you are correct, since in the next equation we truncate the Taylor series anyway. This is done with the usual assumption that we are in the vicinity of a fixed point, since stability is a local property and we are not studying a basin of attraction, this is a reasonable assumption.

---

> ### Author Response · Authors · 2024-11-20
>
> ```
> It seems to me that the model presented in equation 9 can be obtained from the Krotov-Hopfield model by introducing a new variable $\mathbf{y}=\mathbf{Wu}+\mathbf{b}$. Are these two classes of models equivalent under this change of variables, or does the introduction of matrix R make equation 9 more general than Krotov and Hopfield?
> ```
> It is an interesting question. We include additional discussion (lines 408, 409 in the revised version) on that in the main text. The extra matrix $\boldsymbol{R}(\boldsymbol{u})$ does not affect the structure of steady states if it is properly chosen. This means all steady states possible with the old model are also possible with the new model. However, the matrix $\boldsymbol{R}(\boldsymbol{u})$ affects basins of attractions, so they will differ in the new model.

---

> ### Author Response · Authors · 2024-11-20
>
> ```
> In all the derivations $\mathbf{b}$ is assumed to be time-independent. This is fine (same is true in Krotov-Hopfield). I am wondering though, if anything could be said about situations when $\mathbf{b}$ depends on time? E.g., could energy function (10) still guarantee convergences for some properties of $\mathbf{b(t)}$?
> ```
> It is a valid question. Unfortunately, we were unable to get a good answer. All the proofs that we have (as well as all proof of Krotov and Hopfield) are not valid when $\boldsymbol{b}$ is time dependent. In this case energy function should also explicitly depend on time, which is possible in theory, but we were unable to construct one.

---

> ### Author Response · Authors · 2024-11-20
>
> Dear yNbD,
> If you have any additional suggestions after reading the revised manuscript, please do not hesitate to contact us.

---

### Author Response · Authors · 2024-11-20
**Revised manuscript**

We would like to thank reviewers for constructive suggestions. Initial submission is revised in line with reviewers proposals.

This is a short summary of the revision, point-by-point answers are available as individual comments under reviews. For convenience, in the revised version all new material is marked in blue. The main changes introduced are:
1. All proofs are moved to appendices. Proof of Proposition 2 (Appendix B) is substantially extended.
2. Three new appendices unrelated to proof are added:
   1. Appendix A explains in detail how our notation is related to works "Dmitry Krotov and John Hopfield. Large associative memory problem in neurobiology and machine learning", "Dmitry Krotov. Hierarchical associative memory".
   2. Appendix C contains discussion on the situation when energy function is unbounded from below. It also contains a proof that ReLU memory can support a single stable memory vector if energy is bounded from below.
   3. Appendix D explains in detail how our notation is related to work "John J Hopfield. Neurons with graded responses have collective computational properties like those of two-state neurons."
3. A footnote that explains informally what we mean by dead neurons appears in the Introduction.
4. Introduction to Section 2 is carefully rewritten to explain conditions needed for dynamical systems. Here we also point to related work and appropriate Appendix.
5. Figure 2 is added that provides extra explanation on the energy functions we consider in the article. Here we also give examples of other pathological energy functions, discuss one of these cases right away  and provide reference to Abstract C where the other one is described. Section 2 ends with extra clarifications on the same subject.
6. In Section 4 we provide additional explanation of the formal mathematical techniques used later. This explanation will allow the reader to understand the content of this section in simplified terms.

---

### Meta-Review · Area_Chair_zypf · 2024-12-23

**Metareview:**

**Summary:**
This paper studies the Krotov-Hopfield (KH) model of associative memory (Krotov-Hopfield, 2020), and points out that the energy function of their model suffers from the problem of dead neurons, implying that the energy may have flat directions, which in turn may be harmful in investigating stability of the model. This paper then proposes an alternative formulation which uses not the **pre**-activation variables but the **post**-activation variables as the state variables of the model. The proposed formulation is shown to resolve the dead-neuron problem, while maintaining other plausible properties of the KH model.

**Strengths:**
This paper studies the problem of dead neurons in the existing KH model of associative memory, and proposes how to resolve it via adopting the post-activation variables as the state variables of the model instead of the pre-activation variables in the HK model.

**Weaknesses:**
- Most reviewers raised concern about presentation quality and clarity of this paper.
- I noticed several minor errors in the manuscript. See the list below.

**Reasons:**
The reviewers evaluated this paper positively, though some of them expressed concerns about the presentation quality of this paper. Upon my own reading of this paper, I also noticed several points which would require revision: see the following list.

Minor points:
- Page 3, equations (2)-(4): The function $L(y)$ appears without proper description. The fact that $L(y)$ is some convex function of $y$, which defines the activation function $g$ via equation (2) should be explicitly stated.
- Page 3, line 156: $W_21^\top$ → $W_{21}^\top$
- Page 3, line 158: The double indices $ii-1$ and $ii+1$ are not easy to parse. I would prefer $W_{i,i-1}$ and $W_{i,i+1}$. The same applies also to page 10, lines 487 and 497 as well.
- Page 3, line 162: it allowed (the) authors
- page 4, line 165: Authors → The authors
- Page 4, Example 3: The sigmoid function saturates only in the asymptotic where its argument goes to $\pm\infty$, so that the statement in this example would be imprecise.
- Page 4, Example 4: It would be better to use the expression $\tilde{y}(c)$ throughout the example, rather than mixing the expressions $\tilde{y}(c)$ and $\tilde{y}$, in order to make the presentation coherent.
- Page 3, lines 202-204: As mentioned above, hyperbolic tangent and Gaussian saturate only asymptotically.
- Page 4, Proposition 1: $c$ appearing in the definition of the subspace $\mathcal{D}$ should be in boldface.
- Page 5, line 225; page 6, line 302: $VV^\top$ should be replaced with $V(V^\top V)^{-1}V^\top$, because it is **not** assumed that $V$ has orthonormal columns.
- Page 5, line 248: $dg(\rho)$ → $d\rho\,g(\rho)$ (twice)
- Page 5, Proposition 2, item 2: The expression $E(y,\tilde{b})$ does not match the definition (equation (4)) of the energy function $E(y)$.
- Page 6, line 284, equation (6), 413, 428: The second-order terms on the right-hand sides would need the prefactor $\frac{1}{2}$.
- Page 6, line 313: dynamic(s) is stable
- Page 7, line 342: $\sqrt{2(N-k)}$ would require a prefactor.
- Page 8, Proposition 5: The outermost parentheses on the right-hand side of the formula defining $E_1(u)$ are redundant and should be removed.
- Page 8, equation (11): $R^\top F+F^\top R$ → $R^\top F(u)+F(u)^\top R$; The outermost parentheses on the right-hand side of the formula defining $F(u)$ are redundant and should be removed.
- Page 8, lines 406-408, 411: $Wu-b$ → $Wu+b$ (four times)
- Page 8, line 410: gives $F(u)$ → gives $\frac{\partial E}{\partial u}=F(u)f(u)$; $\dot{u}=R(u)f(u(t))$ → $\dot{u}=-R(u)f(u(t))$
- Page 9, line 447: $E=E_1$ → $E=E_2$
- Page 9, Example 7: $erf$ should be in the upright face.
- Page 10, line 497: one observe(s)

**Additional Comments On Reviewer Discussion:**

The concerns on presentation quality and clarity have been addressed by the authors to some extent, which has improved the readability of this paper. After the author rebuttal and the discussion between the authors and the reviewers, the ratings of all the reviewers are on the positive side of the acceptance threshold.

---

> ### Public Comment · ~Vladimir_Fanaskov2 · 2025-02-26
> **Meta Review Change Log Part 1**
>
> We would like to thank the area chair for finding many additional typos. Below is a list of changes we introduce in the response to the received suggestions. In the revisited manuscript (not a camera-ready version), these changes are indicated in red.
>
> ```
> Page 3, equations (2)-(4): The function $L(y)$ appears without proper description. The fact that $L(y)$  is some convex function of $y$, which defines the activation function  via equation (2) should be explicitly stated.
> ```
>
>   In the revised version $L(y)$ is described in lines 134-135. Convexity is not a required property for $E$ to be non-increasing on trajectories.
>
> ```
> Page 3, line 156: $W_{2}1^\top \rightarrow W_{w1}^{\top}$
> ```
>
> This fragment was already corrected in the revised manuscript.
>
> ```
> Page 3, line 158: The double indices $W_{ii-1}$ and $W_{ii+1}$ are not easy to parse. I would prefer $W_{i,i-1}$ and $W_{i,i+1}$. The same applies also to page 10, lines 487 and 497 as well.
> ```
>
> We introduce requested changes to improve readability in both examples the reviewer mentioned.
>
> ```
> Page 3, line 162: it allowed (the) authors
> page 4, line 165: Authors → The authors
> ```
>
> Fixed.
>
> ```
> Page 4, Example 3: The sigmoid function saturates only in the asymptotic where its argument goes to $\pm\infty$, so that the statement in this example would be imprecise.
>
> Page 3, lines 202-204: As mentioned above, hyperbolic tangent and Gaussian saturate only asymptotically.
> ```
>
> While technically the reviewer is correct, practically, the sigmoid function is saturated for finite values of the argument because we are computing with floating point numbers (often with reduced precision).
>
> Also, we want to point out that the imprecise statement is still valid: the fast decay of the sigmoid will prevent us from reliably storing any information in flat regions. Since the one virtue of the distributed representation is robustness, these approximately flat directions still pose the same problem as the exactly flat ones.
>
> We introduce two footnotes making this claim explicit.
>
>
> ```
> Page 4, Example 4: It would be better to use the expression $\widetilde{y}(c)$ throughout the example, rather than mixing the expressions $\widetilde{y}$ and $\widetilde{y}(c)$, in order to make the presentation coherent.
> ```
>
> We replace $\widetilde{y}$ on $\widetilde{y}(c)$ in all expressions in this example.
>
> ```
> Page 4, Proposition 1: $c$ appearing in the definition of the subspace $\mathcal{D}$ should be in boldface.
> ```
>
> We replace $c$ with $\boldsymbol{c}$.
>
> ```
> Page 5, line 225; page 6, line 302: $\boldsymbol{V}\boldsymbol{V}^{\top}$ should be replaced with $\boldsymbol{V}\left(\boldsymbol{V}^{\top}\boldsymbol{V}\right)^{-1}\boldsymbol{V}^{\top}, because it is not assumed that  has orthonormal columns.
> ```
>
> We agree and add a footnote that we assume for simplicity that $\boldsymbol{V}$ has orthonormal columns. This is equivalent to the additional inverse of Gram matrix that the reviewer suggested.

---

> ### Public Comment · ~Vladimir_Fanaskov2 · 2025-02-26
> **Meta Review Change Log Part 2**
>
> ```
> Page 5, line 248: $dg(\rho)\right d\rho g(\rho)$ (twice)
> ```
>
> Fixed.
>
> ```
> Page 5, Proposition 2, item 2: The expression $E(\boldsymbol{y}, \widetilde{\boldsymbol{b}})$ does not match the definition (equation (4)) of the energy function $E(\boldsymbol{y})$.
> ```
>
> We removed the extra argument $\widetilde{\boldsymbol{b}}$ to match the definition.
>
> ```
> Page 6, line 284, equation (6), 413, 428: The second-order terms on the right-hand sides would need the prefactor $\frac{1}{2}$.
> ```
>
> We introduce missing $\frac{1}{2}$.
>
> ```
> Page 6, line 313: dynamic(s) is stable
> ```
>
> Fixed.
>
> ```
> Page 7, line 342: $\sqrt{2(N-k)}$ would require a prefactor.
> ```
>
> We respectfully disagree. In the revised manuscript the proof for this proposition was transferred to the appendix, so we extend the discussion there and show that for a large $N$ spectral radius of the GOE matrix converges to $\sqrt{2N}$. The argument uses theory from Section 4 of https://arxiv.org/abs/1209.3394.
>
> ```
> Page 8, Proposition 5: The outermost parentheses on the right-hand side of $E_1(\boldsymbol{u})$ the formula defining  are redundant and should be removed.
> ```
>
> We removed unnecessary parentheses.
>
> ```
> Page 8, equation (11): $\boldsymbol{R}^{\top}\boldsymbol{F} + \boldsymbol{F}^{\top}\boldsymbol{R}\rightarrow \boldsymbol{R}^{\top}\boldsymbol{F} (\boldsymbol{u}) + \boldsymbol{F}(\boldsymbol{u})^{\top}\boldsymbol{R}$; The outermost parentheses on the right-hand side of the formula defining  are redundant and should be removed.
> ```
>
> We introduce missing arguments and remove redundant parentheses.
>
> ```
> Page 8, lines 406-408, 411: $\boldsymbol{W}\boldsymbol{u} - \boldsymbol{b}\rightarrow \boldsymbol{W}\boldsymbol{u} + \boldsymbol{b}$ (four times)
> ```
>
> We changed the sign from $-$ to $+$ in all four expressions specified by the reviewer.
>
> ```
> Page 8, line 410: gives $F(u)\rightarrow$ gives $\frac{\partial E}{\partial u} f(u)$; $\dot{u} = R(u)f(u(t))\rightarrow \dot{u} = -R(u)f(u(t))$
> ```
>
> We introduce requested changes.
>
> ```
> Page 9, line 447: $E=E_1\rightarrow E=E_2$
> ```
>
> Fixed.
>
> ```
> Page 9, Example 7: $erf$ should be in the upright face.
> ```
>
> We changed \text{erf} to {\sf erf} to change the font in the example environment.
>
> ```
> Page 10, line 497: one observe(s)
> ```
>
> Fixed.

---

### Decision · Program_Chairs · 2025-01-22

Accept (Poster)